# Mixing, Spatial Resolution and Argon Saturation in a Suite of Coupled General Ocean Circulation Biogeochemical Model Configurations off Mauretania

Heiner Dietze[1, 2, *] and Ulrike Löptien[1, *]

[1]Department of Computer Science, Archaeoinformatics - Data Science, University of Kiel, Germany
[2]Department of Chemistry, King's College London, London UK
[*]These authors contributed equally to this work.

**Correspondence:** Heiner Dietze (heiner.dietze@ifg.uni-kiel.de)

**Abstract.** Numerical coupled ocean circulation biogeochemical modules are routinely employed in Earth System Models that provide projections into our warming future to the Intergovernmental Panel on Climate Change (IPCC). Previous studies have shown that a major source of uncertainties in the biogeochemical ocean component is vertical, or rather diapycnal, ocean mixing. The representation of diapycnal mixing in models is affected by several factors, among them are the (poorly
constrained) parameter choices of the background diffusivity, the choice of the underlying advection numerics and the spatial discretization. This study adds to the discussion by exploring these effects in a suite of regional coupled ocean circulation biogeochemical model configurations. The configurations comprise the Atlantic Ocean off Mauretania - a region renown for its complex ocean circulation driven by seasonal wind patterns, coastal upwelling and peculiar mode water eddies featuring toxically low levels of dissolved oxygen. Based on simulated argon saturation as a proxy for effective mixing we conclude
that the resolution effect beyond mesoscale on diapycnal mixing is comparable to other infamous spurious effects, such as the choice of advection numerics or a change of the background diffusivity of less than 60%.

## 1 Introduction

Numerical earth system models (ESMs) based on partial differential equations are generically used to project into our warming future. In contrast to global climate models, ESMs feature an explicit representation of biogeochemical processes that interact
with the physical components of the climate system (e.g. Flato, 2011). This allows for an exploration of the feedbacks between climate and biogeochemical processes in the oceans (and elsewhere) which has been put to use to evaluate ocean-based geo-engineering options (e.g. Keller et al., 2014; Feng et al., 2017, 2020). Even so, for many metrics of societal interest substantial model uncertainties prevail (e.g. Löptien, 2015; Löptien and Dietze, 2017; Löptien et al., 2021).

Among the contemporary efforts to fence-in model uncertainties are the development of techniques that exploit ensembles
of simulations (e.g. Deser et al., 2020) and models (e.g. Counillon et al., 2023): rather than putting trust into one, potentially erroneous model simulation the idea is that an ensemble of similar yet different models and model simulations may cancel out individual model deficiencies and, in addition, can provide a measure of uncertainty by relating to the envelope of all individual

ensemble members. Another route to fencing-in model uncertainties is to identify (e.g. Löptien et al., 2021) and improve the representation of relevant processes in the models (e.g. Rogers et al., 2017; Braghiere et al., 2023). This entails explicitly resolving ever more of the spatial and temporal spectrum as it becomes computationally affordable with the progression of Moore's law (e.g. Gutjahr et al., 2019).

As for the representation of the ocean in ESMs, an infamous uncertainty is vertical mixing (Gnanadesikan et al., 2015; Löptien and Dietze, 2019; Zhu et al., 2020; Semmler et al., 2021; Löptien et al., 2021) which constitutes the primary link between the ocean surface (that is in exchange with the atmosphere) and vast abyssal stocks of nutrients, carbon and heat. Even though vertical mixing is prevalent on rather small spatial (of the order of meters only) and temporal scales (of the order of tens of minutes) it directly impacts the global-scale thermohaline circulation (e.g. Wunsch and Ferrari, 2004; Kuhlbrodt et al., 2007) and effectively controls atmospheric stocks of greenhouse gases (e.g. Schmittner et al., 2009; Sallée et al., 2012; Ellison et al., 2023). This combination of small-scale, short-lived processes and large-scale, long-term compounding effects is a real challenge since the width of spacial and temporal spectra that have to be resolved map onto high computational cost of respective simulations and, further, call for extensive (and expensive) observational surveys. Unfortunately, however, direct microstructure measurements of mixing in the field are labour intensive and difficult to interpret. This has been somewhat alleviated by advances in respective finescale parameterizations which infer mixing "... with built-in averaging over internal wave times and space scales" (Kunze, 2017). But even so, and despite considerable effort (as summarized in e.g. Whalen et al., 2015; Kunze, 2017; Waterhouse et al., 2014; MacKinnon et al., 2017) a comprehensive and reliable 3-dimensional climatology of global oceanic mixing is yet to be compiled. An alternative (to direct measurements), indirect approach is to constrain mixing by fitting general circulation models to available observations in an optimal way; i.e. testing effectively through mixing distributions until an optimal fit to available observations such as temperatures, salinities and pressure is obtained. This approach, however, has been failing so far as, e.g., Trossman et al. (2022) found physical variables to be insufficient to constrain mixing in this way and proposed to include, additionally, biogeochemical variables such as dissolved oxygen. While the latter is technically feasible it can be misleading in those cases where the biogeochemical model formulations that define oxygen consumption (by bacteria) and production (by algae) are imprecise. For those unfortunate but not uncommon cases it has been shown that biogeochemical and physical model deficiencies can compensate one another (Löptien and Dietze, 2019; Ruan et al., 2023).

To date, observational estimates of mixing are sparse and large-scale ocean-circulation simulations rely on mixing parameterizations since the explicit resolution of the entire relevant spectrum of motion is computationally unfeasible. The respective parameterizations range from simple Richardson Number approaches (e.g. Pacanowski and Philander, 2981) to more elaborate closures that include table-lookups based on environmental conditions for flux profiles (e.g. Large et al., 1994) or even explicitly solving equations describing the temporal evolution of turbulence properties (P. Gaspard and Lefevre, 1990; Mellor and Yamada, 1982). Still, the uncertainty in background turbulent diapycnal mixing as applied in ESMs spans an order of magnitude with little to no improvement over the last decade(s) (c.f. Schmittner et al., 2009; Ellison et al., 2023). We speculate that this limited advancement is, in parts, associated to spurious diffusion i.e. the fact that the effective mixing in ESMs is generally not known because spurious effects of advection numerics are substantial but rather unknown contributors.

Early on, Gerdes et al. (1991) illustrated that the specifics of advection numerics in models matter for the simulation of water mass transformation rates in ocean circulation models. This proves the significance of the link between advection numerics and mixing since mass transformation is essentially the result of mixing. Further confirmation came from, e.g., Lee et al. (2002) showing that the spurious numerical mixing of 3-dimensional advection schemes may well be of the order of observational estimates of mixing (and the explicit representation thereof in models Holmes et al., 2021). These findings are well recognized and are the drivers behind ongoing advection scheme development (e.g., McGraw et al., 2024).

In summary, for a given computational cost a compromise has to be made between spurious diffusivity and dispersion of advection numerics. (Somewhat counter to intuition both may ultimately result in spurious diapycnal mixing because the "unmixing" effect of dispersion can cause static instabilities in the water column which, in turn, supplies energy (or a reason) for vertical mixing (schemes) to set in (Hecht, 2010)). The quantification of spurious mixing in realistic (as opposed to idealized) circulation model configurations that apply state-of-the-art advection numerics (as opposed to e.g. simple upstream schemes) is - however - so complex (e.g., Ilicak, 2016) that the essence of the compromise between computational cost and spurious diffusivity is often not known. The core of this problem is the dependency of spurious mixing on local advection velocities and property gradients. Obviously, the advection of a homogeneously distributed property cannot cause spurious mixing (in all sensible numerical implementations of advection). Likewise, advection with zero velocity cannot cause spurious mixing either. For non-zero velocities and property gradients, however, the existence of substantial spurious mixing has been illustrated using clever experimental designs and diagnostics (Griffies et al., 2000; Megann, 2018; Burchard and Rennau, 2008; Getzlaff et al., 2012; Ilicak et al., 2012; Hill et al., 2012; Klingbeil et al., 2014; Gibson et al., 2017; Klingbeil et al., 2019; Banerjee et al., 2023). Among the remaining challenges is to infer spurious mixing at specific locations in space and time (preferably on a time-step and grid-cell base) using a simple, generic and (computationally) cost-effective procedure. (Note, that this challenge is closely related to developing an ideal numerical advection scheme that is devoid of spurious mixing because a successful quantification of spurious effects may be applied to infer respective corrections.)

This study adds to the discussion by exploring the usability of argon saturation as a means or proxy to rank the effective diapycnal mixing (consisting of the sum of both explicitly prescribed and spurious mixing) in members of a suite of regional general ocean circulation models against one another. The idea is to utilise argon saturation as a model diagnostic - as opposed to using the rather sparse argon observations as an additional data constrain along with temperature, salinity and transient tracers. This relatively novel approach exploits the nonlinear relationship between argon saturation in seawater and temperature (and to a lesser extend salinity). It is rooted in findings showing that (1) such non-linear relationships accumulate in the presence of mixing to significant deviations from saturation in the thermocline (e.g. Dietze and Oschlies, 2005a, b) and (2) that this effect can be exploited in order to trace diapycnal mixing both in models and observations (e.g. Henning et al., 2011; Ito et al., 2007; Emerson et al., 2012).

We focus on differences in model configurations as effected by differences in horizontal resolution. The rationale is that the recent years saw substantial progress in that, e.g., the CMIP6 model ensemble (O'Neill et al., 2016) now entails eddy-present and eddy-rich models, capable of resolving mesoscale features in their ocean model components (as reviewed by Hewitt et al., 2020). An examination of respective climate sensitivities is on-going (Ruan et al., 2023, c.f.) but there is consensus that an

eddy-resolving resolution is associated to important model improvements, such as in the representation of boundary currents and ventilation at deep-water formation sites (e.g. Marzocchi et al., 2015). A follow-up question (c.f. Capet et al., 2008; Brett et al., 2023; Karleskind et al., 2011) to which we contribute here, is as to how sub-mesoscale resolving models will differ from contemporary mesoscale resolving models in terms of biogeochemical cycling and air-sea heat exchange. To this end we analyse a suite of regional ocean model configurations in the range between 12 to 1.5 km resolution in the Atlantic Ocean off Mauretania. The region is renown for its complex circulation, biogeochemical processes and fishery (e.g. Doumbouya et al., 2017). Ranking " ... among the four major eastern boundary coastal upwelling systems in the world oceans" (Klenz et al., 2018) it is one of biologically most productive and diverse regions world-wide. As such it is of high ecological and socioeconomic importance and there is concern that a combination of anthropogenic warming, acidification, oxygen depletion and poor resource management may deplete its assets rapidly.

The dynamics in the region of interest are exceptionally diverse which suggests that a multitude of physical and biogeochemical processes that potentially depend on resolution are at play. Off Mauretania, a fertile coastal system intersects with an oligotrophic gyre further offshore. Along the rugged complex coastline (characterized by prominent headlands and bays) coastal currents and nutrient-rich upwelling is driven by seasonal winds (e.g. Siedler et al., 1992; Pelegrí and Benazzouz, 2015; Camp et al., 1991). The seasonality in itself is also diverse in that it increases from North towards the tropics because it is linked to the seasonal migration of the intertropical convergence zone (Messié et al., 2009; Lachkar and Gruber, 2012). Off the coast, in the Cape Verde Frontal System, sub-tropical surface water is mixed with tropical water while traveling west pushed by the North Equatorial Current and north Cape Verde Current, respectively (Mittelstaedt, 1991; Zenk et al., 1991). Evidence of intense meso- and small-scale mixing by eddies has been documented early on in the region (Tomczak, 1978; Tomczak and Hughes, 1980; Barton, 1985, 1987; Mittelstaedt, 1991). These early results were routinely confirmed: For example, von Appen et al. (2020) report on correlations between physical and biogeochemical properties on (sub-mesoscale) spatial scales of several kilometer. Karstensen et al. (2015) find evidence that sub-mesoscale spatial scales are involved in creating so-called "dead-zone eddies" - peculiar mode water eddies containing low (potentially toxic) oxygen concentrations throughout their month-long journey from the coast offshore. In summary, the region of interest is reportedly dynamically diverse and suspected to be affected by small-scale circulation. As such we conclude that the region is well suited to test the effects of going beyond mesoscale resolution in a suite of coupled general ocean circulation biogeochemical models.

The following Section 2 presents the technical aspects of our approach including a description of our suite of model simulations (Section 2.1). Section 2.2 introduced argon as a proxy for mixing, Section 3 compares the model results which are then discussed (Section 3.3) and summarized (Section 4. The Appendix provides a selection of additional comparisons between the model resolutions.

## 2   Materials and Methods

We use four configurations of a *basin-scale*, quasi-realistic, numerical coupled ocean-circulation biogeochemical model that differ from one another in their horizontal resolutions of 12 km, 6 km, 3 km and 1.5 km in a confined region of interest off

Mauretania, respectively (Figure 1, Table 1). All configurations share the same 12 km - resolution bathymetry. The ultra-high 1.5 km resolution configuration has been previously validated favourably in terms of ocean physics with special focus on sub-mesoscale features by Dilmahamod et al. (2021). In the present study we explore differences that emerge as we coarsen the horizontal spatial resolution incrementally such that sub-mesoscale features are no longer resolved. The coarsest model version is additionally integrated with altered background diffusivities and three different advection schemes. The major focus of our

analysis is on diapycnal or diabatic oceanic transports which we trace by introducing the noble gas argon as an additional prognostic tracer.

## 2.1 Coupled Ocean Circulation Biogeochemical Model

All experiments are based on the Modular Ocean Model (MOM), version MOM4p1, (Griffies, 2009) model framework as released by NOAA's Geophysical Fluid Dynamics Laboratory via https://github.com/mom-ocean/MOM4p1, last access 15th

July 2021. The biogeochemical module is BLING - short for **B**iology **Li**ght **I**ron **N**utrients and **G**asses developed by Galbraith et al. (2010). The model configurations are identical to the Southern Ocean setup MOMSO ($\approx 11$ km resolution Dietze et al., 2020) - except for changes related to the grid, bathymetry and forcing. The high-resolution settings build on experience with the 2 km-resolution configuration MOMBA (Dietze et al., 2014) and the 100 m-resolution configuration MOMBE (Dietze and Löptien, 2021).

Even our highest-resolved configuration *High* applies sub-grid scale parameterizations because it cannot resolve the full spectrum of processes (which expands down to centimetre scale). In order to mimic the effect of unresolved vertical turbulent fluxes we apply the k-profile parameterization approach of Large et al. (1994) with a critical bulk Richardson number of 0.3 and a constant vertical background diffusivity and viscosity of $10^{-5}\,m^2\,s^{-1}$ in all of our coupled ocean circulation biogeochemical model simulations (unless explicitly stated otherwise such as in Table 2). These background values apply also below the surface

mixed layer throughout the water column. The choice of background diffusivity is consistent with space and time-averaged estimates of open-ocean conditions derived by Ledwell et al. (1993) (to which substantial uncertainties in our region of interest apply, Schafstall et al., 2010). Both parameterizations of the non-local and the double diffusive (vertical) scalar tracer fluxes are applied. Horizontally, we apply a state-dependent horizontal Smagorinsky viscosity scheme (Griffies and Hallberg, 2000; Smagorinsky, 1993a, b) to keep friction at the minimal level necessitated by numerical stability. The scale of the Smagorinsky

isotropic viscosity is set to a very low 0.01 which effects very low friction and thereby allows for a most vivid circulation. We do not apply any explicit horizontal diffusivity. The (default) advection scheme in the four model versions is the Sweby-advection scheme of Hundsdorfer and Trompert (1994) with Sweby (1984) flux limiters. We dub our reference model setup MOMA: **MO**dular Ocean Model off **MA**uretania. MOMA, in its highest-resolution configuration *High* (see Table 1), has recently been favourably compared to observations by Dilmahamod et al. (2021) with a special focus on sub-mesoscale circulation.

### 2.1.1 Grid and Bathymetry

We use the ETOPO5 bathymetry (Data Announcement 88-MGG-02, Digital relief of the Surface of the Earth. NOAA, National Geophysical Data Center, Boulder, Colorado, 1988). The bathymetry is interpolated onto an Arakawa B (Arakawa and Lamb,

**Table 1.** List of coupled ocean-circulation biogeochemical model configurations. Horizontal resolution refers to both zonal and meridional resolution within respective high-resolution domains (bounded by $30°$ to $15.5°$W and $14°$ to $25°$N).

| name tag | horizontal resolution [$km$] | total number of grid points |
|---|---|---|
| *Coarse* | 12 | $167 \times 168 \times 72$ |
| *CoarseMedium* | 6 | $368 \times 271 \times 72$ |
| *MediumHigh* | 3 | $536 \times 670 \times 72$ |
| *High* | 1.5 | $1427 \times 1159 \times 72$ |

1977) model grid using a bilinear scheme. The model bathymetry is smoothed with a filter similar to the Shapiro filter (Shapiro, 1970). The filter weights are 0.25, 0.5 and 0.25. The filtering procedure can only decrease the bottom depth, i.e. essentially, it

fills rough holes. The filter is applied three times consecutively because we found in other high-resolution model configurations (Dietze et al., 2020, 2014), this to be a good compromise between unnecessary smoothing on the one hand and numerical instabilities introduced by overly steep topography on the other hand.

Our model configurations cover almost the entire North Atlantic Ocean. The boundary is a solid rectangular wall ($15.5°$ to $60°$W and $6.25°$ to $38.5°$N; Figure 1 (a)) that does not allow for any ingoing and outgoing fluxes. The design rationale is to

separate boundaries and the region of interest ($30°$ to $15.5°$W and $14°$ to $25°$N) such that spurious boundary effects fall short of entering the region of interest within our integration time which is few years only. For that reason we chose a model domain that is so much larger than the region of interest.

Within the region of interest which comprises the dynamically complex Cape Verde Frontal System where the North Equatorial Current meets the north Cape Verde Current (Figure 1 (a)) the horizontal resolution is telescoping (Figure 1 (c) and (d))

in all of the configurations as specified in Table 1. The 12 km, 6 km, 3 km and 1.5 km resolution configurations of MOMA are dubbed *Coarse*, *CoarseMedium*, *MediumHigh* and *High*, respectively (Table 1). The vertical discretization comprises a total of 72 levels in all configurations. The resolution is 10 m at the surface, coarsens to 100 m at depth and is cutoff at 1500 m. Figure 1 (a) and (b) show the nominal depth and thickness of each vertical grid level, respectively. The confinement to a maximum of 1500 m water depth facilitates the use of a larger barotropic timestep which reduces computational burden. (Note

that an additional set of simulations, as described in Section 2.1.3, was integrated in order to put the effect of resolution into perspective.)

The computational cost increases from *Coarse* to *High* by more than three orders of magnitude: one order of magnitude increase for each of the increase in zonal and meridional resolution combined with an overhead that is predominantly associated to the smaller time step necessitated by the increased demands in numerical stability as a consequence of higher spatial

resolution. (Higher spatial resolution facilitates the simulation of faster, relatively short-lived dynamics that has to be resolved temporally in order to avoid a build-up of spurious numerical effects.)

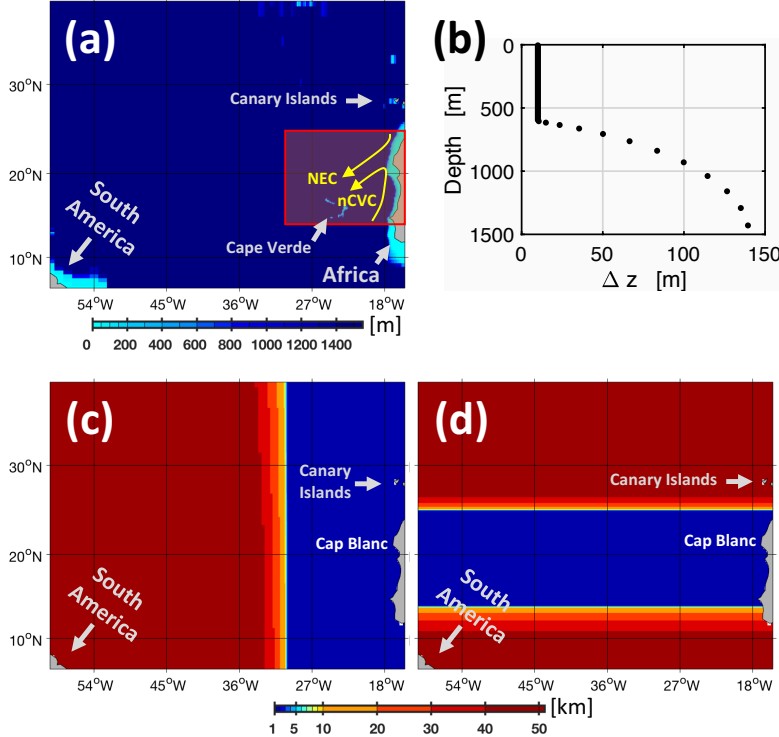

**Figure 1.** MOMA model domain and spatial (finite difference) discretization. Panel (a) shows the model domain and bathymetry. The red rectangle denotes the high-resolution domain of all configurations listed in Table 1. The yellow arrows in panel (a) depict the North Equatorial Current and the north Cape Verde Current. Their convergence defines the Cape Verde Frontal System. Panels (b) shows the vertical discretization of all configuration (as listed in Table 1). Panel (c), and (d) exemplarily show the zonal and meridional resolutions of configuration *High*, respectively.

### 2.1.2 Initial Conditions and Atmospheric Forcing

Initial conditions and forcing are identical to the settings in the Southern Ocean setup MOMSO as documented in the respective model documentation (Dietze et al., 2020).

The configurations are started from rest using climatological annual mean temperatures, salinities, phosphate and oxygen concentrations from Locarnini et al. (2010); Antonov et al. (2009); Garcia et al. (2010a) and Garcia et al. (2010b), respectively. All other biogeochemical prognostic variables (such as dissolved iron concentrations and dissolved organic matter) are initialized with downscaled model output from the global fully spun-up coarse resolution configuration "FMCD" described in Dietze et al. (2017). This pragmatic decision speeds up the equilibration of biogeochemical cycles and saves computing time (but note

that there are more sophisticated approaches to this problem, e.g. Khatiwala, 2024).

As for the atmospheric conditions that (in combination with the simulated surface properties of the ocean such as SST and surface velocities) drive the fluxes of heat, moisture, and momentum onto the ocean we apply the Large and Yeager (2004, 2008) Corrected Normal Year Forcing (COREv2) - a well-tested annual climatological cycle of all the data needed to force an ocean model (note that similar contemporary products exist; Tsujino et al., 2018; Hersbach et al., 2020). The forcing includes (artificial) synoptic variability and air-sea fluxes evolve freely based on bulk-formula. Even so, our approach may well skew the effects of synoptic weather systems which has been shown to be of relevance in regions of strong ocean-atmosphere coupling (e.g. O'Kane et al., 2014). Despite this caveat Dilmahamod et al. (2021) showed that our pragmatic choice of using a climatology in MOMA produces realistic oceanic variability.

### 2.1.3 List of Simulations

The design rationale of our simulation ensemble is to encourage the simulations to diverge from one another in response to the differences in spatial resolution, diffusivities and the choice of the advection schemes: We start from rest such that all configurations are shocked by the sudden onset of the atmospheric forcing which excites waves and non-linear circulation patterns that are free to develop independently on the respective spatial model grids. Further, we encourage diverging air-sea heat and momentum fluxes by applying non-linear bulk formulas based on the simulated oceanic surface properties in the respective calculations.

We compare simulations that have not reached equilibrium yet (after one year of spin-up and a relatively short overall integration time of a maximum of three years) which limits potentially spurious effects of our lateral boundary conditions (c.f. Appendix B) and saves computational resources. Note in this context that Dilmahamod et al. (2021) showed that the circulation of MOMA is already realistic after a spin-up of only one seasonal cycle. Consistent with this result we find that the energy content of our eddy-field does not fall short of observational estimates after one year already (Figure 5). This suggests that our simulation period of 3 years is long enough to reveal major effects of differing resolution (even though it may take much longer to equilibrate potential energy).

We integrate two sets of model simulations. The first set features 4 differing resolutions as listed in Table 1. The simulations are dubbed *Coarse*, *CoarseMedium*, *MediumHigh* and *High*.

The second set of simulations is targeted at putting the differences among the first set into perspective. The focus is on effective domain-averaged diapycnal mixing between *Coarse* and *High* as traced by simulated argon saturation. The simulations explore a range of (uncertain) parameter settings for the vertical background mixing/diffusivity (*Coarse 1.2*, *Coarse 10*) and numerical advection schemes (*Coarse upwind, Coarse QUICKer, Coarse MDPPM*) (Section 2.2.1, Table 2).

### 2.2 Model Analyses

We focus on ranking the effective diapycnal mixing in our simulations against one another. Effective, diapycnal mixing is infamously difficult to quantify in models (Griffies et al., 2000; Megann, 2018; Burchard and Rennau, 2008; Getzlaff et al., 2012). Here we add to the discussion by introducing dissolved argon as an additional prognostic tracer to our simulations in order to track mixing by respective oversaturation.

**Table 2.** Simulated dissolved argon over-saturation in units % on nominal October 1st 1902 averaged over depth (0 to 400 m) and the respective high-resolution model domains (bounded by 30° to 15.5°W and 14° to 25°N). Rates refer to nominal year 1901.

| name tag | description | $\Delta Ar[\%]$ | $\frac{\Delta Ar}{time}[\frac{\%}{yr}]$ |
|---|---|---|---|
| *High* | 1.5 km horizontal resolution; $1 \times 10^{-5}\,m^2s^{-1}$ background diffusivity | 0.43 | 0.14 |
| *Coarse* | 12 km horizontal resolution; $1 \times 10^{-5}\,m^2s^{-1}$ background diffusivity | 0.45 | 0.17 |
| *Coarse 1.2* | 12 km horizontal resolution; $1.2 \times 10^{-5}\,m^2s^{-1}$ background diffusivity | 0.45 | 0.16 |
| *Coarse 1.4* | 12 km horizontal resolution; $1.4 \times 10^{-5}\,m^2s^{-1}$ background diffusivity | 0.46 | 0.16 |
| *Coarse 1.6* | 12 km horizontal resolution; $1.6 \times 10^{-5}\,m^2s^{-1}$ background diffusivity | 0.47 | 0.16 |
| *Coarse 1.8* | 12 km horizontal resolution; $1.8 \times 10^{-5}\,m^2s^{-1}$ background diffusivity | 0.47 | 0.17 |
| *Coarse 2* | 12 km horizontal resolution; $2 \times 10^{-5}\,m^2s^{-1}$ background diffusivity | 0.50 | 0.16 |
| *Coarse 10* | 12 km horizontal resolution; $1 \times 10^{-4}\,m^2s^{-1}$ background diffusivity | 0.77 | 0.25 |
| *Coarse upwind* | 12 km horizontal resolution; upwind advection scheme | 1.15 | 0.37 |
| *Coarse QUICKer* | 12 km horizontal resolution; QUICKer advection scheme (Leonard, 1979) | 0.44 | 0.17 |
| *Coarse MDPPM* | 12 km horizontal resolution; resolution with piece-wise parabolic method (Griffies, 2009) | 0.41 | 0.15 |

### 2.2.1 Argon Saturation - a Proxy for Mixing

As suggested by Dietze and Oschlies (2005a) we employ an explicit representation of argon in our suite of configurations in order to trace the effect of mixing: argon (Ar) is a noble gas and, once isolated from the sea surface, is a conservative tracer in the ocean (e.g. Bieri et al., 1966). Argon oversaturation, $\Delta Ar$ is defined as:

$$\Delta Ar = \frac{Ar}{Ar_{sat}} \times 100\% - 100\%, \tag{1}$$

where $Ar$ and $Ar_{sat}$ denotes the actual and saturated Argon concentrations, respectively. In contrast to concentration the
saturation is not conservative: a mixed water-parcel will always carry a higher argon saturation than the arithmetic mean of the original parcels since the saturation curves are convex over the entire range of oceanic temperatures (and salinities). Hence, an increase in $\Delta Ar\%$ may be indicative of mixing (while negative values can be indicative of spurious dispersion).

Even though there is consensus on the link between mixing and $\Delta Ar$ (Dietze and Oschlies, 2005a; Ito and Deutsch, 2006; Henning et al., 2011; Ito et al., 2007; Emerson et al., 2012; Hamme et al., 2017; Getzlaff et al., 2019) a ubiquitous relationship
between standing stocks of argon saturation and diapycnal mixing that would allow for a global, localized assessment of effective diffusivities in numerical models has not been established yet. Among the challenges yet to overcome is to correct for processes other than mixing that affect $\Delta Ar$ such as subsurface warming by penetrating solar radiation (e.g. Dietze and Oschlies, 2005b) and bubble entrainment (e.g. Spitzer and Jenkins, 1989; Hamme et al., 2017).

In this study a ubiquitous relationship is not targeted. Instead, we aim to use $\Delta Ar$ as a relative measure to rank our simulations
with respect mixing against one another. To this end our simulated argon concentrations are more of a simple analytical tool designed to compare simulations among one another rather than a metric capable of constraining the absolute effective mixing in models with observations (such as developed by e.g. Holmes et al., 2021). This facilitates the implementation because we

can initialize all simulations with $\Delta Ar = 0\%$ and are not bound to integrate until the saturations have equilibrated because we only study the differences between our simulations rather than linking simulated concentrations to observations.

## 245  3  Results

A comprehensive and favourable comparison between simulation *High* and observations is provided by Dilmahamod et al. (2021). Here we focus on illustrating that our simulation *High* is capable of resolving sub-mesoscale features. Further, we will employ $\Delta Ar$ to compare simulations with respect to the effect of diapycnal mixing.

### 3.1  Resolving Mesoscale and Sub-Mesoscale Dynamics

Figure 2 shows a sea surface temperature snapshot of the respective entire model domain after 2 years, 3 months and 11 days of spinup. The simulated sea surface temperatures look very similar both outside and inside the respective high-resolution domain. Differences between the resolutions become only apparent at extreme zoom levels such as shown in Figure 3 for an upwelling filament off Cap Blanc. On these scales it is evident that a higher spatial model resolution clearly improves the representation of small circulation features such as the relatively warm SST feature near 18°W, 21°N that is completely absent in Coarse.

Figure 4 confirms the visual impression of increasing levels of detail with increasing resolution by showing respective vertical relative vorticities, calculated as:

$$\zeta^z = \partial_x v - \partial_y u \tag{2}$$

normalized by the Coriolis frequency $f$. Given that sub-mesoscale structures are associated to values of magnitude $O(1)$ (e.g. Shcherbina et al., 2013), Figure 4 shows that the configurations *Coarse* and *CoarseMedium* are devoid of sub-mesoscale dy-
namics, while *MediumHigh* features sporadic occurrences and *High* ubiquitous coverage. This warrants a comparison between sub-mesoscale and coarser resolution using our configurations *Coarse* and *High*.

Figure 5 show eddy kinetic energy (EKE) averaged meridionally over the respective high-resolution model domain as observed from space and simulated with *High* and *Coarse*. As to be expected, the eddy kinetic energy increases with higher spatial resolution because the sub-mesoscale circulation starts to contribute to the integrated energy spectrum. Hence, simulation *High*
features higher EKE-levels than *Coarse* (also because the energy cascade from large to smaller spatial scales is apparently not accelerated by the increase in resolution). Both EKEs in *High* and *Coarse* do not fall short of observational estimates (neither during the upwelling nor during the relaxation season). This implies that the simulations capture enough dynamics for a meaningful comparison of the differing configurations (as, reversely, a comparison of *High* and *Coarse* with overly sluggish circulations would be inconclusive). On a side note we add, that Dilmahamod et al. (2021) finds no evidence for excessive
overestimation of our simulated eddy dynamics in *High* while, at the same time, there is evidence that eddy kinetic energy estimates from space are biased low (Fratantoni (e.g. 2001); but see also Luecke et al. (2020) for agreements between models and observations across frequency bands).

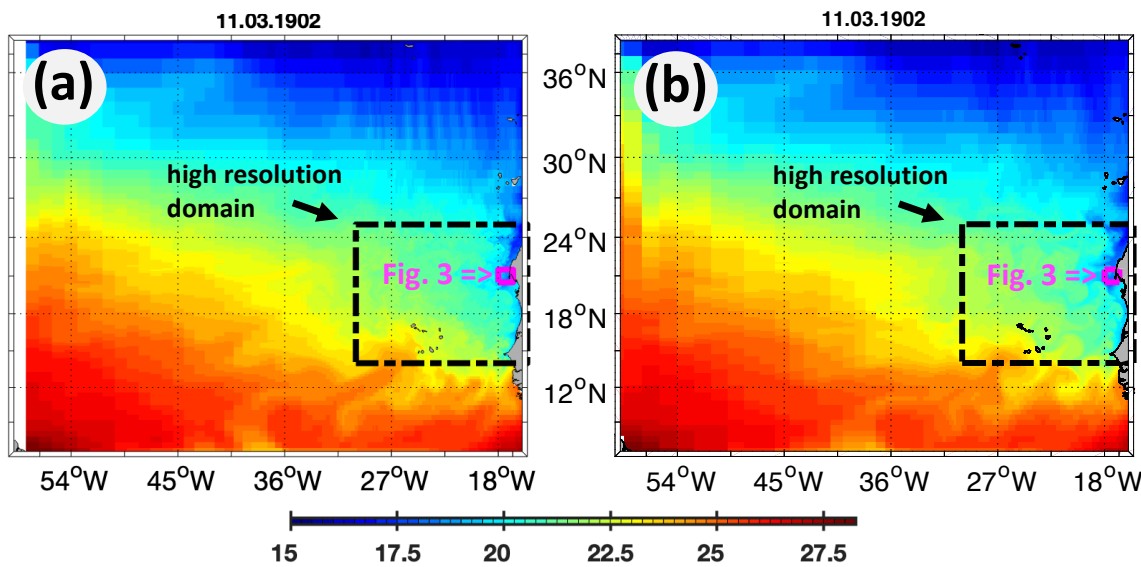

**Figure 2.** Exemplary snapshots of simulated sea surface temperatures in units degrees centigrade. Panel (a) and (b) feature the entire model domain of the high-resolution configuration *High* and the coarse resolution configuration *Coarse*, respectively. The black dashed line denotes the region where both zonal and meridional resolution telescopes (Figure 1 ) as specified in Table 1. The tiny magenta squares off Cap Blanc at the Coast of Mauretania depicts the boundaries of the zoom shown in Figure 3.

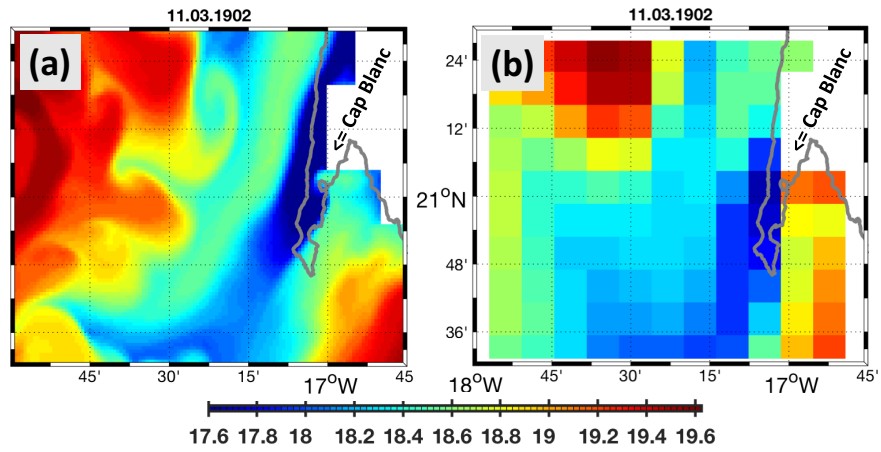

**Figure 3.** Exemplary snapshots of simulated sea surface temperatures capturing an upwelling filament off Cap Blanc. Panel (a) and (b) show the high-resolution configuration *High* and the coarse resolution configuration *Coarse*, respectively. Note that the entire model domains are larger than shown here (Figure 2).

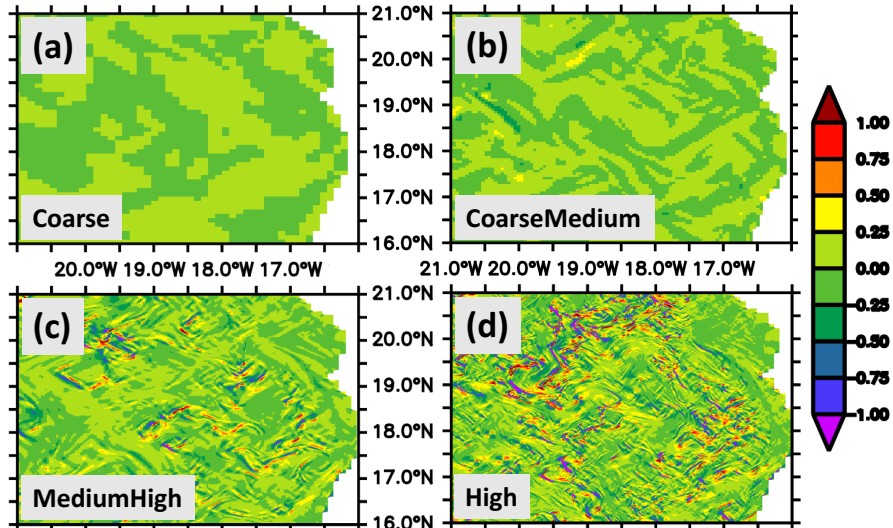

**Figure 4.** Exemplary snapshot of simulated vorticity during the upwelling period (on nominal 15 January 1902). The vorticity is calculated from surface currents averaged over the upper 50 m and normalized by the Coriolis frequency. Panel (a), (b), (c) and (d) showcase the effect of horizontal resolution in our configurations. Relative vorticity of magnitude $O(1)$ indicate sub-mesoscale flows. The region shown corresponds to $\approx 15\%$ of the area of the high-resolution model domain.

Small-scale mesoscale and sub-mesoscale features have been associated with locally-intensified vertical fluxes (e.g. Lévy et al., 2001; Mahadevan and Archer, 2000; Mahadevan and Tandon, 2006; Martin and Pondaven, 2003). Consequently, it
has been argued that vertical exchange should increase along with spatial resolution since ever more small-scale features are resolved with increasing resolution. One of the processes potentially leveraging this effect is surface current wind interaction (Martin and Richards, 2001): Ekman pumping is calculated from the curl of the wind stress. The resolution of ever smaller spatial scales in ocean currents yields higher spatial derivatives of wind stress, calculated from the squared difference of wind speed and oceanic surface current velocity. Hence, simulated Ekman pumping, associated with the wind stress curl, increases
with model resolution. Table 3 confirms that higher resolution yields higher Ekman velocities in our configurations. This applies for both the mean of the magnitude and the standard deviation which approximately double from *Coarse* to *High*. Figure 6 showcases the signature of ocean circulation in the Ekman pumping in more detail. In *Coarse* the prevalent structure is boxlike with a spatial scale of $O(100\,km)$ effected by the scale of the (spatially discretised) atmospheric forcing. Increasing resolution to *CoarseMedium*, *MediumHigh* and *High* results in ever more structure, apparently related to an eddying circulation
with local values increasing up to several meters per day. Hence we expect, consistent with the pioneering work of, e.g., Lévy et al. (2001); Mahadevan and Archer (2000); Mahadevan and Tandon (2006); Martin and Pondaven (2003), an increase in vertical, fluxes of heat, salt and biogeochemical entities with increasing resolution in our configurations. We are particularly curious whether the increase in Ekman dynamics with increasing resolution leads to enhanced diapycnal mixing or if its effect is primarily advective.

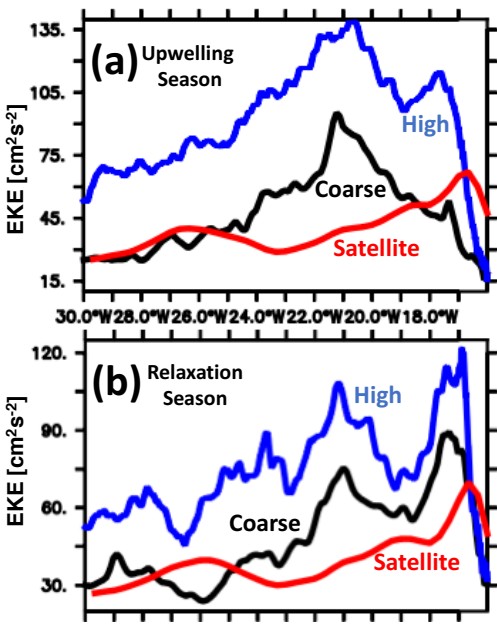

**Figure 5.** Eddy kinetic energy, domain averaged over the high-resolution model domain ($30°$W - $15°$W and $17°$N - $24°$N). Panel (a) and (b) refer to the Upwelling Season (December - April) and the Relaxation Season (May - July), respectively. The red line refers to observations from space (Aviso 1993-2010). The black and blue lines denote model output averaged over nominal years 1901-1902 simulated with configuration *High* and *Coarse*, respectively.

**Table 3.** Ekman pumping as simulated within the high-resolution model domains on nominal 15th January 1901 in units $m\,day^{-1}$.

| tag | mean of magnitude | standard deviation |
|---|---|---|
| *Coarse* | 0.16 | 0.20 |
| *CoarseMedium* | 0.18 | 0.23 |
| *MediumHigh* | 0.21 | 0.29 |
| *High* | 0.28 | 0.41 |

## 3.2 Ocean Mixing

Based on previous work (e.g. Lévy et al., 2001; Mahadevan and Archer, 2000; Mahadevan and Tandon, 2006; Martin and Pondaven, 2003) we expect an increase in vertical mixing in response to increasing resolution of intense and localised up- and downwelling events. Alternatively, we may find that the effect of such events cancel out on larger scales in our model (Reissmann et al., 2009; Eden and Dietze, 2009; Dietze and Löptien, 2016).

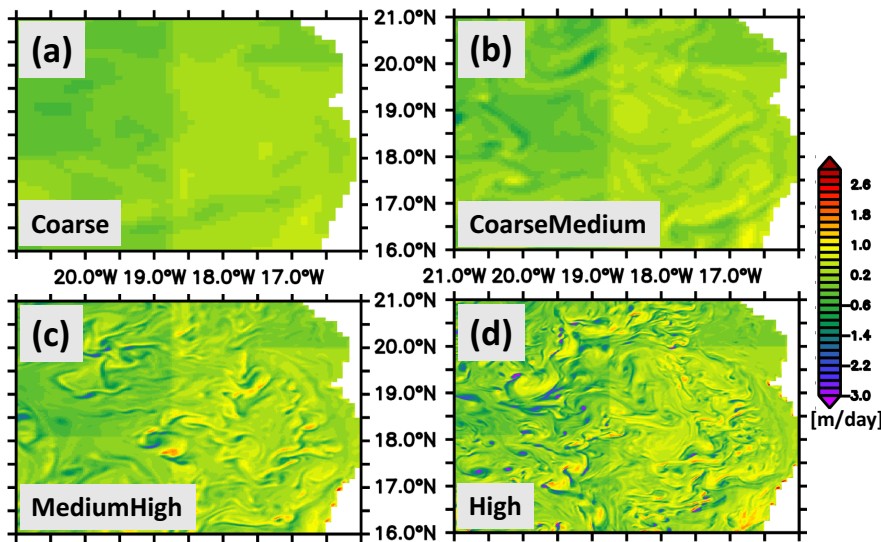

**Figure 6.** Exemplary snapshot of simulated Ekman pumping (nominal 15 January 1901). The Ekman pumping includes surface current wind effects. Panel (a), (b), (c), and (d) refer to configuration *Coarse*, *CoarseMedium*, *MediumHigh* and *High*, respectively. The region shown corresponds to ≈ 15% of the area of the high-resolution model domain.

In this Section, we use argon over-saturation as a proxy recording the mixing history of water parcels since these were last in contact with equilibrating air-sea fluxes at the sea surface. We start with an illustration of effects, continue with comparing the different spacial resolutions and end with putting the latter into perspective by comparing them to other simulations.

    The interpretation of argon saturation is not straightforward because it is antagonistically affected by mixing: Diapycnal mixing produces oversaturation as explained in Section 2.2.1. At the same time however, mixing transports oversaturated water
to the surface where it loses its oversaturation to the atmosphere. Figure 7 (a) shows a Hovmöller diagram of the argon saturation averaged over the high-resolution domain which illustrates the antagonistic processes at work. Above the surface mixed layer the water is always saturated because it is subject to equilibrating air-sea fluxes. Below the surface mixed layer the water is shielded from the direct effect of air-sea fluxes and three effects prevail in our simulations: (1) Solar radiation penetrating below the surface mixed layer increases the temperature in summer. This reduces the solubility and increases the saturation
(e.g. Dietze and Oschlies, 2005b). (2) Mixing of water parcels (effected by the sum of explicitly prescribed diffusivity and implicit spurious mixing from numerical schemes) with differing temperatures and salinities results in oversaturation because solubility is a nonlinear function of temperature and salinity (e.g. Dietze and Oschlies, 2005a; Ito and Deutsch, 2006; Emerson et al., 2012). (3) Mixing with equilibrated surface water across the surface mixed layer and abyssal water still carrying their initial 0% oversaturation. In combination these effects drive a seasonal saturation maximum situated directly below the surface
mixed layer and a linearly downward protruding saturation signal over time.

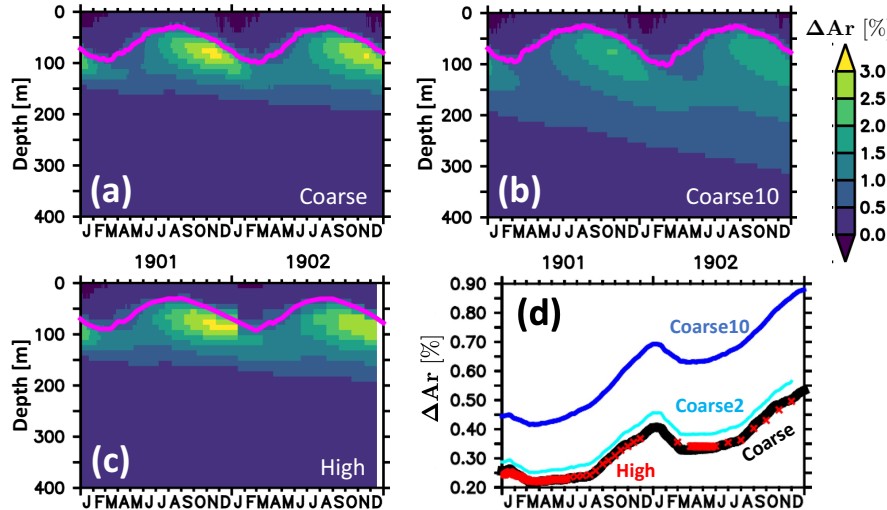

**Figure 7.** Simulated argon oversaturation in units % averaged over the high-resolution model domain ($30^\circ$W - $15^\circ$W and $17^\circ$N - $24^\circ$N). Panel (a), (b), and (c) refer to model configurations *Coarse*, *Coarse10* (featuring a ten-fold increase in background diffusivity relative to *Coarse*) and *High*, respectively. The magenta line denotes the surface mixed layer defined with a $\Delta = 0.5\,K$ temperature criterion. Panel (d) shows corresponding oversaturation averaged over 100 to 400 m depth for configurations *High*, *Coarse*, *Coarse2* and *Coarse10* . *Coarse2* refers to a simulation with a two-fold increase (from 1 to $2 \times 10^{-5}\,m^2\,s^{-1}$) in background diffusivity.

A comparison between Figure 7 (a) and (b), showing results from *Coarse* and *Coarse10* (Table 2), respectively, reveals how an (exaggerated) 10-fold increase in explicitly prescribed background diffusivity manifests itself: the higher diffusivity smoothes the maximum below the surface mixed layer by distributing the saturation signal vertically. Part of the signal enters the surface mixed layer and is eradicated by equilibrating air-sea fluxes while part of it is protruding downwards. When

computing water column averages below the surface we see in Figure 7 (d) that *Coarse10* accumulates deep oversaturation at a faster pace than *Coarse* - even though its losses to the ventilated surface mixed layer are greater due to enhanced vertical exchange by mixing. Note, that this applies also when averaging over the entire water column (not shown). A comparison between *High*, *Coarse* and *Coarse10* in Figure 7 suggests that the effective diapycnal mixing in *High* is not larger than in *Coarse* because *High* features a similar or rather smaller increase relative to *Coarse*: near the end of the simulation the averaged

$\Delta Ar$ in the 100 to 400 m depth range amounts to 0.43% in *High* relative to 0.45% in *Coarse*. Also, the visual impression of the simulated argon saturation for *Coarse* and *High* in Figure 7 (a) and (c) is rather, while (b) is distinctly different.

Table 2 and Figure 9 puts the small differences in simulated averaged oversaturation values between *High* and *Coarse* into perspective: From a suite of simulations with background diffusivities 1, 1.2, 1.4, 1.6, 1.8, 2 and $10 \times 10^{-5} m^2 s^{-1}$ we conclude that our method can reliably detect differences in background diffusivity down to $0.4 \times 10^{-5} m^2 s^{-1}$, but fails at distinguishing

very low diffusivities - probably because the implicit numerical diffusion is then masking the effects of our choices of explicit diffusivity.

Overall we conclude that the difference between *High* and *Coarse* in diffusivity as detected by argon saturation is rather minor and comparable to other infamous and contemporary uncertainties such as switching from one to another state-of-the-art advection scheme (Table 2; default scheme in *Coarse* versus *Coarse QUICKer* and *Coarse MDPPM*). Note in this context, that

the effect of using the outdated upwind scheme introduces much larger effects on the diapycnal mixing and this effect is even stronger than a 10-fold increase *Coarse 10* in background diffusivity.

### 3.3  Discussion

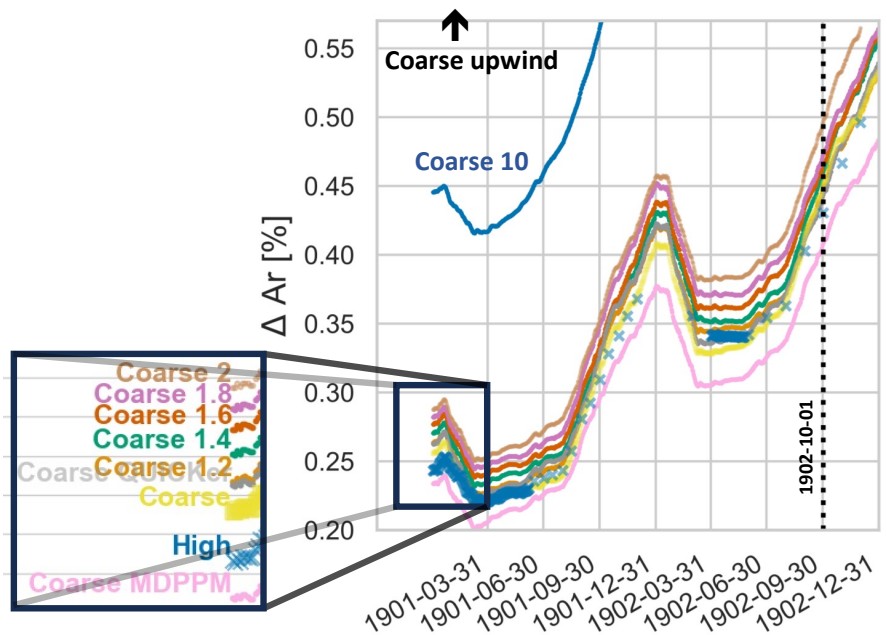

**Figure 8.** Temporal evolution of simulated argon oversaturation in units % averaged over the high-resolution model domain (30°W - 15°W and 17°N - 24°N) and 100 to 400 m depth starting after the first year of spinup. Data gaps such as in configuration *High* are attributable to I/O-failures. The black-dashed line marks nominal October 1902 which is the base of Table 2. Configuration *Coarse upwind* is off the scale. Model tags are explained in Table 2.

Our study exploits the nonlinear, convex relationship between argon saturation and temperature (and salinity) in order to rank effective diapycnal mixing in a suite of model simulations. Our range of calibration simulations where the background

diffusivity was varied between $1 \times 10^{-5} \, m^2 s^{-1}$ and $1 \times 10^{-4} \, m^2 s^{-1}$ show that simulations with differences in background diffusivities of more than $4 \times 10^{-6} \, m^2 s^{-1}$ are consistently different in that higher diffusivities are associated with higher saturations throughout year two and three of the simulation (Figure 8, 7, Table 2).

Clearly, the saturation signals due to mixing are low in amplitude. This obstructs a comparison of our model output with real world observations where additional antagonistic processes such as bubble entrainment and effects of finite gas exchange

that can fail to keep up with the cooling effect on saturation in convective regimes (e.g. Seltzer et al., 2023) are at play. The representation of these effects is - to-date - dependent on model formulations which is work in progress (e.g. Pṗppelmeier et al., 2023); driven by an increasing observational base and interest in general (e.g. Jenkins et al., 2023; Hamme et al., 2017). Nonetheless, our results show that our method consistently ranks simulations with varying background diffusivity and advection schemes, suggesting robustness of argon saturation as a comparative metric. So how does it compare to other

contemporary approaches to diagnose diffusivities in model simulations?

    The main advantage of our method is its simplicity. Typically all models provide an interface to add a passive tracer so that our implementation of argon to trace mixing reduces to merely calculating initial conditions (i.e. 100% saturation based on inital temperatures and salinites) and setting of a surface boundary condition where argon is reset to 100% saturation every timestep. This sets it apart from more sophisticated approaches based on variance decay (e.g. Burchard and Rennau, 2008;

Schlichting et al., 2023) or available potential energy (e.g. Ilicak, 2016).

    Our approach based on argon saturation is apparently similar to the approach of Holmes et al. (2021) that is based on diffusively effected heat fluxes: First, we find that their diffusively effected heat fluxes yield qualitatively similar patterns to our estimates of "diffusively effected" argon fluxes (Figure A1 of Holmes et al. (2021) versus Figure 3 in Dietze and Oschlies (2005a)). Second, we report a detection accuracy of down to $4 \times 10^{-6}\, m^2 s^{-1}$ which is similar to what we derive from Figure 17

in Holmes et al. (2021) where their simulations 025-NG0, 025-KBV and 025KB5 showcase that their approach can detect changes in diffusivities down to $5 \times 10^{-6}\, m^2 s^{-1}$.

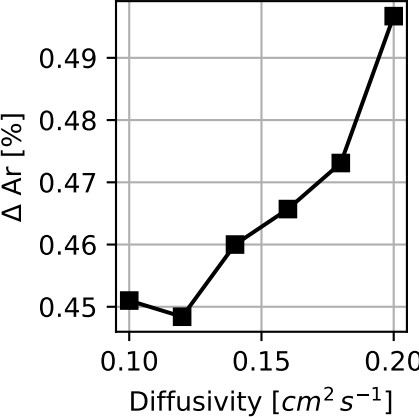

**Figure 9.** Simulated argon oversaturation as a function of explicit background diffusivity. Each value corresponds to a *Coarse* simulation with respective diffusivity for temperature, salinity and Ar. The argon oversaturation is calculated as averaged horizontally over the high-resolution model domain (30°W - 15°W and 17°N - 24°N) and vertically over 100 to 400 m depth. For reference, simulation *High* features 0.43% (Table 2).

## 4   Conclusions

We implemented an explicit, prognostic representation of dissolved argon into an ocean-circulation biogeochemical model of the North Atlantic. A suite of calibration simulations with systematically altered background diffusivities established a link between argon saturation and effective vertical diffusivity. This link was employed to rank effective diffusivity in a suite of model configurations with differing spatial resolutions off Mauretania against one another. Further, we tested different numerical advection schemes in order to set the resolution effects into perspective.

Our results suggest rather modest effects, comparable to contemporary numerical errors when switching from a $12\,\mathrm{km}$ mesoscale to a $1.5\,\mathrm{km}$ sub-mesoscale resolution. More specifically we find the effect on argon based on this change in resolution is comparable to a change to another contemporary advection scheme or to a change of the background diffusivity of less than 60% (note that all these minor effects appear below the detection limit of our method while larger changes in diffusivity could be clearly identified).

Our results are is in-line with recent findings of Brett et al. (2023); Karleskind et al. (2011) who reported on surprisingly large similarities between sub-mesoscale resolving simulations and the same simulations with an artificially damped circulation (Porcupine Abyssal Plain and POMME region, respectively).

Caveats apply. Among them are (1) the application of the hydrostatic approximation (which we applied in line with contemporary IPCC models but in contrast to e.g. Mahadevan and Archer (2000)), (2) the effects of a combined increase in horizontal and vertical resolution (e.g. Stewart et al., 2017) has not been tested, (3) our results apply for the region off Mauretania only and (4) potentially accumulating long-term effects of differing resolutions have not been studied.

## Appendix A:  Generic Model Output

The main body of the paper is focussed on effective vertical mixing as traced by argon saturation rather than showcasing results from the suite of simulations. Here we present more generic model output such as simulated air-sea fluxes, export production and simulated dissolved oxygen. We focus on averages covering the respective highly-resolved subdomains rather than going for side-by-side comparisons between snapshots. The reason being that eddying model simulations are known to show a highly nonlinear behaviour that can manifest itself in notable difference even in repetitions of the exact same experiment using identical executables and hardware (e.g. Dietze et al., 2014, "computational uncertainty" in their Figure 21). Further, detailed differences between individual circulation features may be considered irrelevant in the context of IPCC-type modelling - if they average out on the larger spatial scales that are assertive to coupled components such as the atmosphere.

Among the oceanic peculiarities in the region of interest is an exceptionally complex seasonally varying coastal circulation including deep-reaching undercurrents. Figure A1 shows an example as resolved in exemplary snapshots by configuration *High* and *Coarse*: during the upwelling season, at $18°N$, *High* features a double-cored poleward undercurrent consistent with observations (Dilmahamod et al., 2021, their Figure 2). Configuration *Coarse* is similar but features overall weaker currents. This makes it less realistic in terms of its representation of the undercurrent and more realistic in terms of its representation of the near-surface equatorward current on the shelf. During the relaxation season an intensification of wind stress curl induces an intensification of poleward flow. This, along with ebbed-away equatorward surface current, is reproduced by both *High* and *Coarse*. In summary, *Coarse* is strikingly similar to *High* in terms of seasonality and spatial structure of near-coastal currents, except for resolving less small-scale details.

These similarities of complex seasonal coastal dynamics may, despite the fact that they appear very small, still allow for substantially differing distributions of heat and salt due to the nonlinear nature of the system. This may be of relevance especially for those surface properties that couple the ocean with the atmosphere such as sea surface temperature (SST). Table A1 illustrates, however, that domain-averaged sea surface temperatures, sea surface salinities and near surface nutrients differ also very little across the model configurations. Further, these differences are not all monotonically aligned with the spatial model resolution which suggests other reason for the (very small) differences between the configurations.

Figure A2 provides evidence that the small SST differences do not amplify to considerable air-sea heat flux differences: the exemplary snapshots after almost three years of spinup are - irrespective of resolution - strikingly similar with the only difference being the amount of small-scale structure resolved (which may, or may not be associated with Ekman pumping driven by the surface-current wind effects described above). This is consistent with the rather low sensitivity of effective diffusivity as traced by argon saturation in Section 3.

In the following our focus is on domain-averaged persistent effects that may be missed by coarse resolution configurations rather than on local small-scale transient phenomena: Figure A3 (a) shows the ensemble mean domain-averaged seasonal heat flux. In line with results from Faye et al. (2015) and Foltz et al. (2013) we find a substantial seasonality ranging from a $200\,W\,m^{-2}$ oceanic heat loss in winter up to a $100\,W\,m^{-2}$ heat gain in summer. Figure A3(b) shows that the difference among the configurations is of the order of few $W\,m^{-2}$, corresponding to deviations of several percent only. These differences are

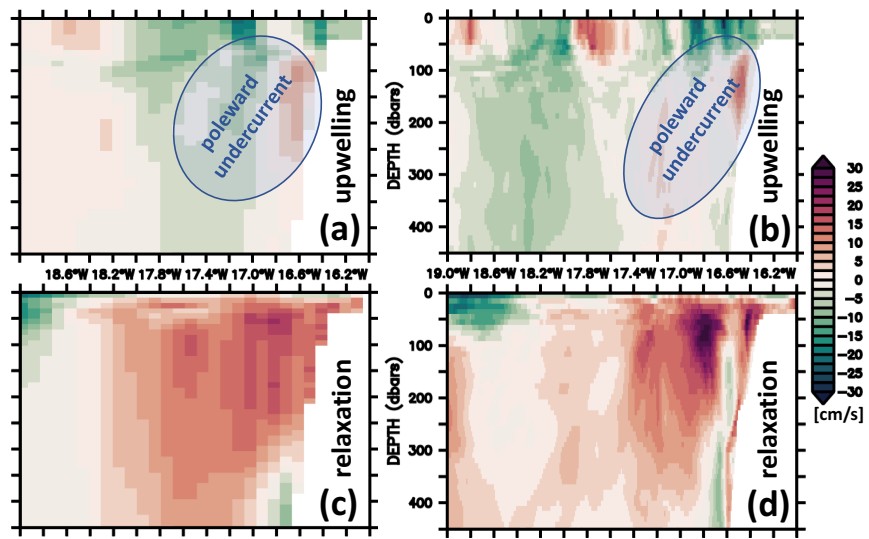

**Figure A1.** Simulated alongshore currents for configurations *Coarse* (a, c) and *High* (b, d) along a zonal section at $18°$N. Panels (a) and (b) refer to exemplary snapshots during the upwelling season (on nominal 1 February 1902) while (c) and (d) refer to the relaxation season (on nominal 1 June 1901).

**Table A1.** Sea surface temperature (SST), sea surface salinity (SSS) and phosphate concentration simulated with respective model configurations (resolutions). All values are averaged over the respective high-resolution model domains (bounded by $30°$ to $15.5°$W and $14°$ to $25°$N) at the end of the simulation on nominal 31st of December 1902. Phosphate concentrations refer to vertical averages over the euphotic zone (0 - 120 m depth).

| Configuration | SST $[°C]$ | SSS [PSU] | PO$_4$ $[mmol\,P\,m^{-3}]$ |
|---|---|---|---|
| *Coarse* | 23.67 | 36.15 | 0.040 |
| *CoarseMedium* | 23.66 | 36.15 | 0.057 |
| *MediumHigh* | 23.69 | 36.11 | 0.039 |
| *High* | 23.63 | 36.11 | 0.042 |

well within the uncertainty range even for observational estimates (e.g. Gulev et al., 2007). In addition we report, again, that
there is no apparent consistent relationship between the deviation from the ensemble mean and the spatial model resolution.

Similar conclusions apply to simulated biogeochemical entities: the vertical gradients of temperature and nutrients are typically opposing one another with warm, sun-lit surface waters being depleted in nutrients (essential for phytoplankton growth) and abyssal, cold waters being nutrient replete. Enhanced vertical transports are expected to increase oceanic heat gain by exposing additional cold water to the relatively warm atmosphere (in our region of interest). Similarly, enhanced vertical transport
result in an increase of nutrient availability to phytoplankton at the sun-lit surface which should manifest itself in e.g. increased

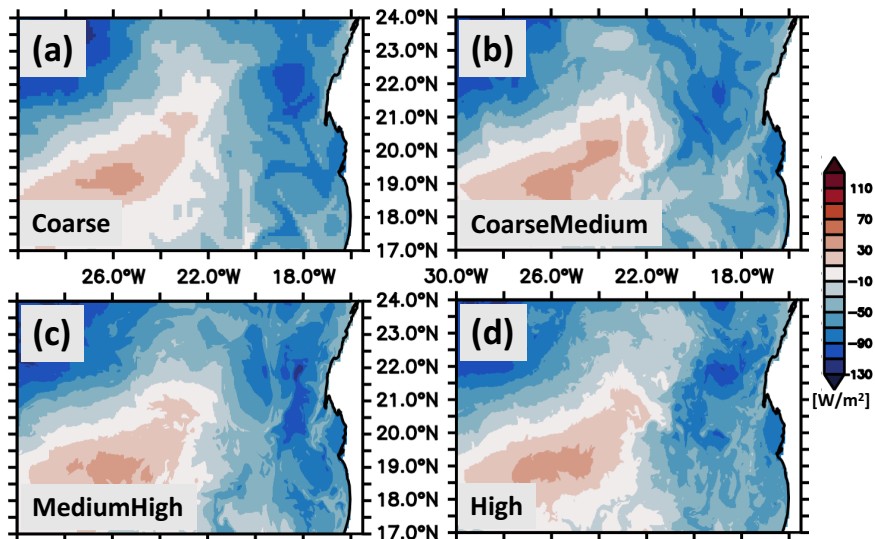

**Figure A2.** Snapshot of simulated daily mean sea-air heat flux (nominal 17 December 1902). Panel (a), (b), (c) and (d) refers to configuration *Coarse*, *CoarseMedium*, *MediumHigh* and *High*, respectively. Positive numbers denote oceanic heat gain.

biotic export production. Figure A4 (a) shows the ensemble mean domain-averaged export production over the coarse of the year with peaking values between 40 and $50\,mmol\,P\,m^{-2}\,yr^{-1}$ during the spring bloom and wintertime values of only several $mmol\,P\,m^{-2}\,yr^{-1}$. Throughout this strong seasonal cycle the individual configurations deviate only a few $mmol\,P\,m^{-2}\,yr^{-1}$ from the ensemble mean and, again, there is no consistent relationship between deviation and resolution (Figure A4(b)).

### A1   Dead-Zone Eddies - a Case Study

Among the regional peculiarities off Mauretania are anticyclonic mode water eddies (ACMEs) that travel from the shelf break hundreds of miles offshore carrying a body of low-oxygenated water (Karstensen et al., 2015). Scientific interest in these 'dead-zone eddies' includes, e.g., the sensitivity of organisms to oxygen deficit (Hauss et al., 2016). Here, we present them as episodic events benchmarking the effect of resolution. Our interest is rooted in that ACMEs are highly nonlinear events shaped by an interplay of complex ocean circulation with biogeochemical cycles. Our rationale is that this nonlinearity could amplify the rather insignificant differences in metrics discussed so far. To this end, the following dead-zone eddy case study is designed to find differences between the resolutions that may be of relevance to other components of the earth system such as higher trophic levels and fisheries.

Figure A5 shows a respective comparison between *Coarse* and *High*. Panel (a) and (b) start with showcasing an anticyclonic mode water eddy (ACME) being shed off the Mauretanian shelf. An analyses of underlying dynamics is provided by Dilma-hamod et al. (2021) and a more comprehensive visualization of processes is animated at https://nextcloud.ifg.uni-kiel.de/index. php/s/SxKm2fKSpaTd7Y2. Both *High* and *Coarse* feature a pronounced seasonal cycle of dissolved oxygen concentration on

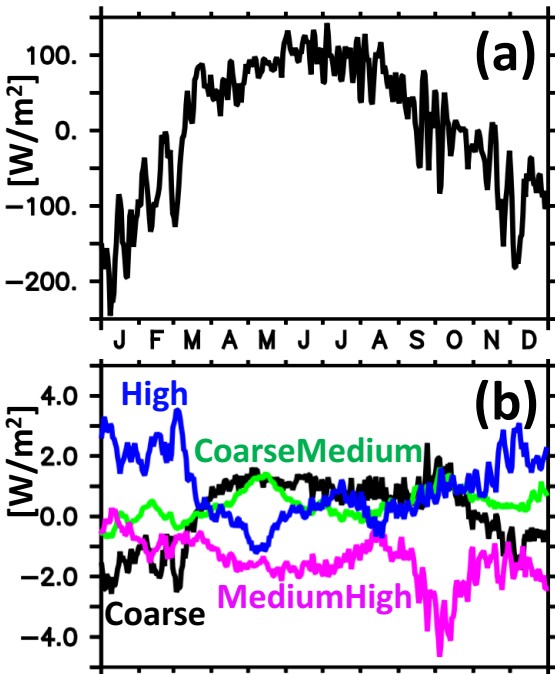

**Figure A3.** Simulated mean sea-air heat flux averaged over the high-resolution model domain (30°W - 15°W and 17°N - 24°N) (nominal year 1902). Panel (a) shows the ensemble mean of configuration *Coarse*, *CoarseMedium*, *MediumHigh* and *High*. Positive numbers denote oceanic heat gain. Panel (b) shows respective differences to the ensemble mean.

the shelf, which is consistent with results from Mittelstaedt (1991): during the upwelling season the shelf is sheltered from relatively cool and oxygen rich upwelling waters offshore. The biotic production on the shelf is exceptionally high (e. g. Wolff
et al., 1993) and this fuels declining levels of oxygen at depth. In line with Mittelstaedt (1991) we find that the isolation of shelf water is permeated by occasional export events. Figure A5(a) and (b) show such export events in configuration *High* and *Coarse*, respectively: ACME seeded with low-oxygenated shelf water are formed and travel offshore for months (Figure A5 (c) and (d)) retaining their low-oxygenated signature.

The upwelling season ends in July south of Cape Blanc (Mittelstaedt, 1991, their Figure 4) when the winds weaken and
veer into a more on-shore direction driving currents that flush the shelf with oxygenated water of off-shore origin (ould Dedah, 1993). The oxygen on the shelf is then replenished until the cycle restarts in spring. Figure A5 (e) and (f) show that the ACMEs are associated to a biotic export production that is elevated relative to ambient waters in our simulations. Even so, both configurations retain rather constant oxygen concentrations within the ACMEs' cores. We conclude that both configurations *High* and *Coarse*, simulate a series of complex coupled ocean circulation biogeochemical processes driving the evolution of
oxygen in ACMEs in a surprisingly similar manner. Apparently, the differences among the configurations are limited to details rather than to materially different processes and dynamics.

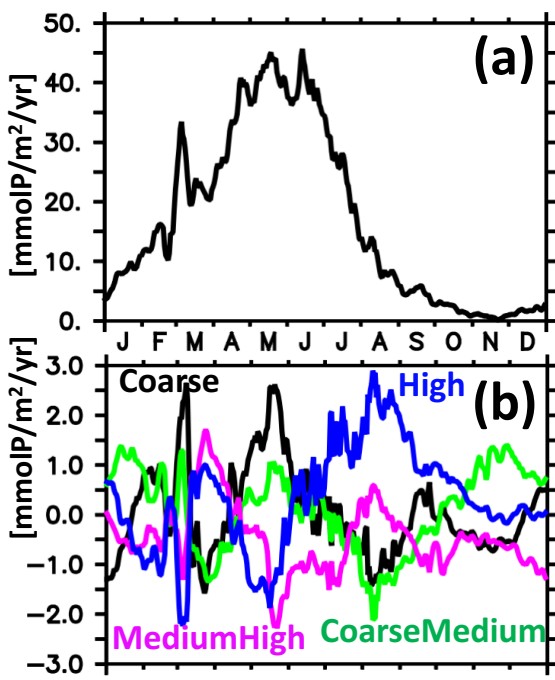

**Figure A4.** Simulated export production calculated as net biotic phosphate sources and sinks integrated over the upper 150 m water column and averaged over the high-resolution model domain (30°W - 15°W and 17°N - 24°N) (nominal year 1902). Panel (a) shows the ensemble mean of configuration *Coarse*, *CoarseMedium*, *MediumHigh* and *High*. Panel (b) shows respective differences to the ensemble mean.

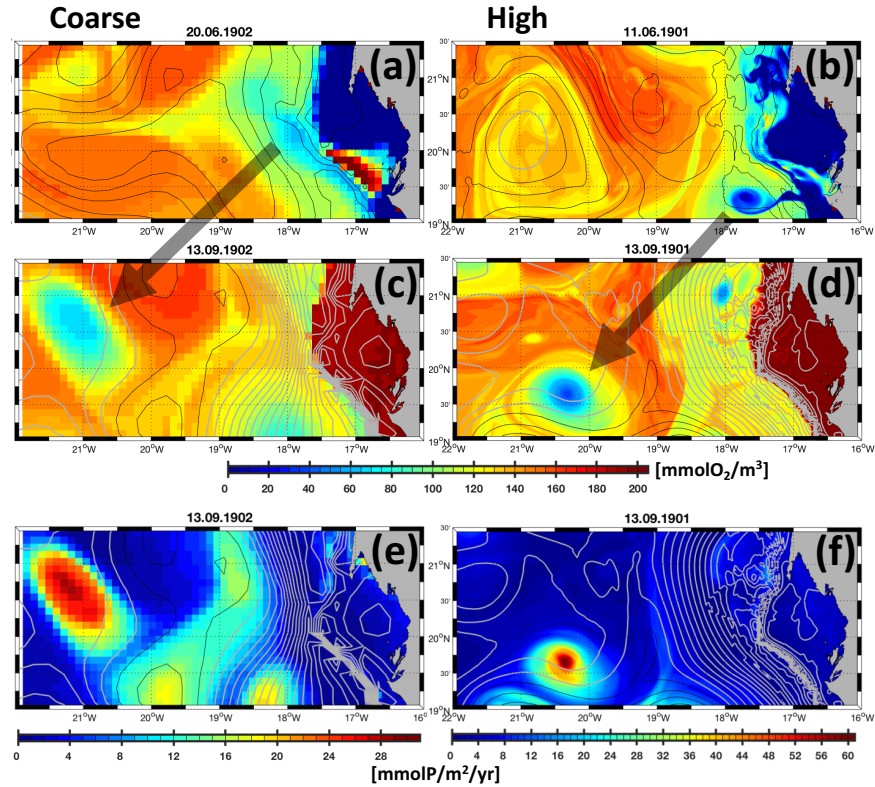

**Figure A5.** Simulated spawning and westward progression of an hypoxic anticyclonic mode water eddy (ACME). Panels (a), (c), (e) refer to configuration *Coarse* and (b), (d), (f) to *High*. The color in panel (a), (b), (c) and (d) denotes the minimum oxygen concentrations found locally in vertical water columns. Panel (e) and (f) show color coded export production. Grey and black contours denote positive and negative sea surface height anomalies spaced at $2\,cm$. Panel (a) and (b) feature the ACME spawning in June as simulated with configuration *Coarse* and *High*, respectively. The grey arrows pointing into Panel (c) and (d) indicate the ACMEs' westward progression until September simulated with configuration *Coarse* and *High*, respectively. Panel (e) and (f) show export production calculated as net biotic phosphate sources and sinks integrated over the upper $150\,m$ water column in September from configuration *Coarse* and *High*, respectively. The respective maximum export production coincides with the position of the ACMEs.

## Appendix B: Boundary Conditions

The model configurations used in this study are peculiar in that the Northern, Southern and Western boundaries are closed walls rather than a realistic representation of the polar gyre, the equatorial current system and the coast of America, respectively. Figure B provides information as to how this may affect the model results in the high-resolution model domain: By introducing an artificial tracer into configuration *Coarse* that is continueosly reset to zero at these boundaries and counts up time elsewhere we count the time elapsed since water parcels have been in contact with the boundaries. After an integration of 30 years we find that water in the 100 m to 400 m depth interval entering the high-resolution model domain is older than three years (Figure B1). This suggests that our findings at the end of our three year-long simulation are rather unaffected by our simple representation of boundaries.

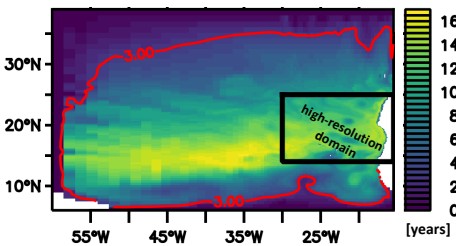

**Figure B1.** Simulated time elapsed since water parcels have been in contact with the Northern, Southern or Western boundary of the model domain (monthly mean at the end of a 30 year-long integration of configuration "Coarse"). The time elapsed shown is averaged over the 100m to 400m depth interval. The red contour depicts the 3-year isoline indicating that the contoured region is rather undisturbed by boundary effects at the end of a 3 year-long simulation.

*Code and data availability.* Model output and additional visualizations are available at https://zenodo.org/records/5549894. The model configurations are available upon request.

*Author contributions.* HD and UL have been equally involved in setting up and running the model configurations. Both authors contributed to the interpretation of model results and to outlining and writing the paper in equal shares.

*Competing interests.* The authors declare that they have no conflict of interest.

*Acknowledgements.* We acknowledge discussions with Ahmad Fehmi Dilmahamod, Julia Getzlaff and Johannes Karstensen. We are grateful to Steven Griffies and the MOM community for sharing their work. Eric Galbraith is acknowledged for sharing BLING. This is a contribution to DFG project number 491008639 "Projecting critical coastal oxygen deficits by the example of the Eckernförde Bight". The work and effort of anonymous reviewers greatly improved the quality of this paper. Thank you!

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
