# Peer review of "Mixing, Spatial Resolution and Argon Saturation in a Suite of Coupled General Ocean Circulation Biogeochemical Model Configurations off Mauretania"

_EGUsphere, 2024_

## Author Comment (AC1)

**Answers to Reviewer #1 (manuscript #: 2024-918)**

"Argon Saturation in a Suite of Coupled General Ocean Circulation Biogeochemical Models off Mauretania" by H. Dietze and U.Löptien"

Point-by-point response. The original reviewer's comments are in regular font, our respective responses are in *italic*.

with the

R: General comments:

The authors of this manuscript examine the model-simulated inaccuracies in diapycnal mixing that cannot be explained by the horizontal resolution near Mauretania and find that it's comparable to advection numerics and choices of background diffusivity by making use of argon saturation as a proxy for effective mixing. I can appreciate how much time and effort that went into this work but I have some reservations about publishing this manuscript as is. One concern that I have is that there is no comparison of the effective diapycnal mixing with other methods, like that of Holmes et al. (2021) or potentially separating out the spurious contribution to compare with a method like that developed in Ilıcak (2016)'s "Quantifying spatial distribution of spurious mixing in ocean models". I understand the authors just want to have a relative measure to rank their model configurations with their effective mixing against one another but without knowing whether their method is properly quantifying the effective mixing, the ranking could be misleading. For example, the authors include the mixed layer in their calculations of the effective mixing, despite how there are known confounding effects on the effective mixing. The second issue is that the authors try to connect Ekman pumping/suction to effective diapycnal mixing without explaining the relationship between the diapycnal advection and diapycnal mixing in the Ekman layer. The third issue is that the model spin-up time may suffice for many applications of model experiments, but they are investigating a signal that is so small that the rate at which their model diagnostics are changing at the time they are evaluating them may be relevant to the significance of the small differences they're finding. I suggest major revisions. Specific comments are listed below:

*A: We appreciate the reviewer's time and effort to improve our manuscript and are grateful for the constructive comments! In summary, we agree with the reviewer who asks for a more convincing presentation of the capabilities of our new method along with a more comprehensive embedment of our work in that of others. We feel that the reviewer is right and will update our manuscripts accordingly.*

*More specifically we find the reviewer's "issues 1 and 3" are essentially rooted in the same shortcoming of our original manuscript. As we understand, the perception of the reviewer is the following: (1) we present a new method to rank the effect of vertical diffusivity/diapycnal mixing, (2) then we find that our method is affected antagonistically by diffusivity, (3) we run the model/method only for a short period that appears random in length to the reader; such that the reader cannot rule out that the effects we discuss are also random - in the sense that we may have picked a time where the relation is coincidentally in favor of our method/reasoning and finally, (4) we do not make a good job to put the capabilities of our approach into the context of what is already out there.*

*Based on these comments we realized that we did not explain the simulations that were designed to test and calibrate the method (where we prescribe different background diffusivities) well enough, and also that we need to present some more analysis to be convincing. Following the suggestions of the reviewer (below) we think that these issues can be well addressed by adding more analysis focused on the temporal evolution of Ar saturation of the simulations with deliberately altered background diffusivity. For the revised version of the manuscript, we plan analysis alike the following Figure 1 which shows the temporal evolution of simulated argon saturation anomalies, where we subtracted the simulated argon saturation of the reference simulation from respective simulations with increased background diffusivities:*

[Figure]

[Figure]

*Figure 1: Temporal evolution of simulated Ar saturation anomaly for different explicit background diffusivities averaged vertically and horizontally over 30° to 15.5°W and 14° to 25°N). The blue line (plus 20% diffusivity) refers to the difference between simulation Coarse 1.2 and the reference simulation Coarse (as defined in Table 2 of the original manuscript). The green, red and black line refer to respective differences of Coarse 1.8, Coarse 2 and Coarse 10 to the reference simulation Coarse (dubbed plus 80%, plus 100% and plus 1000% diffusivity). The left (right) panel excludes (included) the maximum surface mixed layer.*

*The blue (green, red and black line) in Figure 1 corresponds to the difference in Ar saturation between a simulation with an explicit diffusivity increase of 20% (80%, 100%, 1000%) and the coarse resolution reference model version. We find that if we exclude the (maximum winter) surface mixed layer (as suggested by the reviewer – thank you!) all diffusivities rank consistently throughout time, with the higher diffusivities being associated to higher Ar anomalies. By including the entire water column (as in the original version of the manuscript) the same applies with the one exception being that the simulations with an 80% and 100% increase in the background diffusivity come occasionally very close to one another. From the latter we conclude that the accuracy of the ranking struggles when the differences in the prescribed background diffusivities are too small (i.e. when they differ only by 20% or less relative to the diffusivity of the reference simulation). Hence our method appears well suited to rank according diffusivity for differences larger than 20% relative to the reference value (which corresponds to a "detection accuracy" of ~2x10$^{-6}$ m$^2$/s). So how does this compare to other contemporary approaches?*

*Following the reviewer's suggestions (thank you for your constructive literature suggestions!) we will put our method and its detection accuracy into the context of the works of Holmes et al. (2021) and Ilıcak (2016) in the revised version of the manuscript. In a nutshell: The approach of Ilicak (2016) is essentially based on available potential energy – a concept that is yet to be tested in a model with realistic topographic features such as ours. In contrast, the method of Holmes et al. (2021) has been proven to be applicable in "realistic" ocean model configurations such as ours. Unfortunately, their method needs quite some coding and is (to us) not trivial to implement since it must run "online" taking track of every model timestep (which is also recommended for the approach of Schlichting et al. (2023)). We agree heartedly, that a comprehensive comparison of the approaches is warranted and we will contact the authors accordingly hoping that we will get something like a diffusivity diagnostic intercomparison project on the way. (Hopefully, such a project will also include isopycnal models).*

*For the time being we will discuss the following in the revised version of the manuscript:*

*The Holmes (2021) assessment of diffusivity by calculating "diffusively effected" heat fluxes yields qualitatively similar patterns to our estimates of "diffusively affected" argon fluxes (Figure A1 of Holmes (2021) vs. Figure 3 in Dietze and Oschlies (2005)).*

*In terms of "detection accuracy" of Holmes (2021) we find their comparison of simulations 025-NG0 (0m$^2$/s explicit background diffusivity), 025-kbv (5x10$^{-6}$ m$^2$/s) and 025-kb5 (10$^{-5}$ m$^2$/s) in their Figure 17 most instructing as it shows that their method is capable of "resolving" (i.e. successfully ranking one simulation over another) differences in background diffusivity of 5x10$^{-6}$ m$^2$/s. For our method and region, we were able to "resolve" ~2x10$^{-6}$ which suggests that their method and ours are comparable (although more research is clearly needed because it is a bit apples and oranges i.e. different resolutions, regions and integration times are compared). We will end the discussion of this topic in the revised version manuscript with stating that a direct comparison of the methods (hopefully including a variance decay approach as of Schlichting et al. (2023)) in a set of benchmarking experiments is warranted.*

*As concerns "issue 2": the Ekman pumping has been put forward in the literature to make the point that resolution matters in terms of vertical (and eventually diapycnal) transports. We realize (and agree with the reviewer), however, that such a relation is neither obvious nor stringent. We will (also depending on the other reviewer comments) either add a more comprehensive discussion on the link between Ekman pumping and diapycnal transports, or tone down on Eckman pumping, or both in the revised version of the manuscript.*

***Literature:***
*Dietze and Oschlies (2005) https://doi.org/10.1029/2004JC002453*
*Holmes et al. (2021) https://doi.org/10.1029/2020MS002333*
*Ilicak (2016) https://doi.org/10.1016/j.ocemod.2016.11.002*
*Schlichting et al. (2023) https://doi.org/10.1029/2022MS003380*

R: Specific comments:

Line 3: The authors say "unacquainted vertical" by which they are referring to "diapycnal" but that way of describing diapycnal sounds more literary than it sounds like it has specificity; did the authors use AI to help them write this abstract?

*A: Being non-native speakers, we rely heavily on the online dictionary [www.leo.org](www.leo.org) to translate from our mother tongue. It has been around quite a while and we think it does not (yet?) use AI. Anyways, we apologize for alienating wording.*

R: Lines 38-39: The canonical reference for uncertainties in representation of diapycnal mixing (not just vertical) is MacKinnon et al. (2017)'s "Climate Process Team on Internal-Wave Driven Ocean Mixing" even though there's also shear-driven and other mixing

*A: Thank you! We will include this reference.*

R: Line 45: This seems like more "coupled" reasoning than circular because the biogeochemical modules can potentially be developed to be more realistic without developing the diapycnal diffusivity modules, but of course the biogeochemical variables depend upon the diapycnal diffusivities. It's difficult to point to new model innovations in ESMs that don't have this coupling problem. Your argument seems analogous to saying that assimilating sea-ice concentrations with the goal of improving the sea surface properties is circular reasoning because our sea-ice modules are imperfect. In any case, without improvements in the biogeochemical modules, the information propagated by the assimilation of biogeochemical variables may not inform the diapycnal diffusivities as much as the biogeochemical parameters themselves, which is the problem you're pointing out. This would certainly be the case with the Green's function-based approach of ECCO-Darwin (see Carroll et al. (2020)'s "The ECCO-Darwin Data-assimilative Global Ocean Biogeochemistry Model: Estimates of Seasonal to Multi-decadal Surface Ocean pCO2 and Air-sea CO2 Flux") because the physical variables aren't changed in their formulation; only the biogeochemical parameters are altered. I believe the proposal by Trossman et al. (2022) is that the ECCO-Darwin and ECCO setups could be used in an iterative process to incrementally improve the biogeochemical parameters and the physical parameters, respectively, but this is even more computationally expensive than the current process and may require a different process for sequential data assimilation systems, which they present unique issues with in their appendix. Additional evidence that this could work comes from Skakala et al. (2022)'s "The impact of ocean biogeochemistry on physics and its consequences for modelling shelf seas," where it was shown that assimilation of a suite of biogeochemical variables had an impact on the diapycnal diffusivities beyond that of physical variables.

*A: This discussion is really at the heart of what we try to figure out eventually (with our entire body of work)! We really appreciate your thoughts (and view of the respective literature) and will include them (not only in this paper).*

R: Lines 63-64: Will this always be the case? The atmospheric community seems to have come up with a solution to advection numerics in McGraw et al. (2024)'s "Preserving Tracer Correlations in Moment-Based Atmospheric Transport Models". Also, is it possible that spurious diapycnal mixing is the result of spurious vertical advection sometimes? This certainly happens when you perform sequential assimilation (see: Pilo et al. (2018)'s "Impact of data assimilation on vertical velocities in an eddy resolving ocean model").

*A: Not sure - and yes, maybe we should be more optimistic there. Also certainly there has already been a huge progress in the development of advections schemes since the early days of ocean modelling (cf. the upwind scheme relative to modern schemes in our Tab.2). In any case, thanks for the McGraw reference! We will definitely include it in the refined version of the manuscript as it, to the very least, proves that it is still an active field of research. As to*

*the applicability of the approach in the ocean where we have a lot of lateral boundaries (topography) we are not so sure (i.e. we are in the process of understanding the scheme).*

*And yes, spurious diapycnal mixing can definitely be the result of spurious vertical advection. We will add this discussion to the revised version of the manuscript.*

R: Lines 81-82: Isn't there also the effect of surface disequilibrium at the time of water mass formation (e.g., when subduction occurs) that needs to be accounted for? Both the nonlinearity of solubility in the presence of mixing and surface disequilibrium effect cause apparent oxygen utilization to be different from true oxygen utilization, for example. So the argon tracer needs to be injected below the mixed layer such that it doesn't get obducted throughout the experiment. Also, argon can't be injected into the ocean to derive diapycnal diffusivities from observations (unless the background levels are well-known) because there are significant background levels of argon in the ocean, which will eventually render it another exotic tracer on top of the ones Ledwell used. I mention this because it was noted that argon could be used as a proxy for effective diapycnal mixing in both models and observations in the text.

*A: Agreed. Disequilibrium effected by bubble entrainment, finite air-sea gas exchange, subsurface solar heating and ice formation/melting are potentially retarding the usability of Ar as a proxy for mixing in real-world observations (as opposed to being applied to diagnose mixing in numerical models). We will add a respective discussion pointing this clearly out.*

R: Line 150: I understand not wanting to compete with Jason Momoa in Google searches by dubbing your model's acronym MOMOA, but you're going to compete with the Museum of Modern Art in New York City instead. In any case, the model and its tracer conservation properties won't be as much of a problem as they would be if the authors chose another model like HYCOM. And the model domain is reasonable, considering problems with boundary conditions that can occur.

*A: Thanks for the encouragement. The reasoning behind all our ocean model configuration acronyms (MOMBA https://gmd.copernicus.org/articles/7/1713/2014/, MOMBE https://bg.copernicus.org/articles/18/4243/2021/, MOMSO https://gmd.copernicus.org/articles/13/71/2020/) is to make it very clear that the all the heavy lifting (i.e. writing the actual computer code which, literally, is as long as the Bible) has been done by the **M**odular **O**cean **M**odel community and not by us. Also, the MOMA model has already been published under this name.*

R: Line 169: You say that there are 55 geopotential levels in all model configurations here but in Table 1, it says there are 72 grid points in an undefined dimension. The caption of Table 1 only mentions zonal and meridional directions but there is a third dimension listed for the number of grid points. Is that dimension time or is one of your instances of vertical dimensions inaccurate (i.e., in Table 1 or on Line 169)?

*A: Thank you for catching this! Ln. 169 is in error. There are 72 vertical boxes.*

R: Line 186: For future equilibrations of biogeochemical variables, you could consider using anther method like the one by Khatiwala (2024)'s "Efficient spin-up of Earth System Models using sequence acceleration"

*A: True. We will add the respective idea and paper to the revised version of the manuscript.*

R" Line 189: I would argue that use of COREv2 forcing is no longer considered standard, as JRA55-do has supplanted COREv2 in OMIP2 and ERA5 is also commonly used

*A: Thank you! Very true. We will change the wording here to make sure that readers are not misguided and rather use the new forcings in future simulations.*

R: Lines 201-203: If you're interested in the evolution of each of the model configurations, then you should be assessing metrics that are related to the time rate of change, not averages

*A: Agreed, we will add respective information in the revised version of the manuscript, i.e. additional numbers will supplement the accumulated changes over the three-year period.*

R: Lines 205-206: One year of spin-up is common amongst published high-resolution model simulations (based on the global kinetic energy metric you seem to be using) because the initial shock is essentially adjusted away but the model's climatological state is still being approached asymptotically. That would need at least several more years to achieve. It may take much longer to equilibrate potential energy, which includes available potential energy relevant to the eddying circulation, but to first order using kinetic energy may be good enough.

*A: We will add this information to our revised version of the manuscript.*

R: Table 2: Continuing on from my previous comments, please include the time rate of change (linear slope of a regression against time, for example) of the Delta Ar[%] metric you include the average of in this table. That would help the reader see whether these values are close enough to what they would be in a more well-spun-up state. Also, why are you including the mixed layer in your calculations when Argon is not a conservative tracer near the sea surface and diapycnal mixing is not very meaningful within a bulk mixed layer? Are you not using a bulk mixed layer?

*A: This makes sense to us! We will include the time rate of change for each of the consecutive years. We envision to add something like Figure 1 of this rebuttal (which shows the temporal evolution and addresses the "including mixed-layer" vs. "not-including-mixed-layer" issue).*

R: Lines 234-236: Again, because penetrating solar radiation, bubble entrainment, and possibly other processes affect Delta Ar[%], why are you including the mixed layer in your calculations?

*A: You are right. By discarding the upper water column down to the maximum winter surface mixed layer we actually got rid of a lot of noise (mainly the effect of subsurface solar heating accumulating below the summer surface mixed layer). We will take your advice and recalculate accordingly (see Figure 1 above for a first cut).*

R: Lines 266-268: This is qualitatively accurate. However, it can be noted that there are in-situ estimates of eddy kinetic energy compared with high-resolution model simulations in the literature, like those presented in Luecke et al. (2020)'s "Statistical Comparisons of Temperature Variance and Kinetic Energy in Global Ocean Models and Observations:

Results From Mesoscale to Internal Wave Frequencies". Also, along-track SWOT data resolves the higher wavenumbers and there are preliminary results that much more of the spectrum is resolved with SWOT. I'm not suggesting that you perform a comparison with in-situ (e.g., moored) or SWOT data. This is just something to be aware of.

*A: Thanks. We will add the respective references to the revised version of the manuscript.*

*R:* Figure 3: It's curious that the relatively warm SST feature near 18W, 21N in the High configuration is completely absent in Coarse, except for what seems like a separate merging feature near the upper part of the domain plotted here. The rest of the figure comparison looks like the High configuration's SST was coarse-grained.

 A: *We agree that the high and coarse-resolution setups look very similar except for details (such as those next to the coastline). We will highlight the near coastal differences in the revised manuscript.*

R: Table 3 and Figure 6: Okay, the higher the resolution, the greater the Ekman pumping/suction there is in its mean and variability, but how come you're spending time on this when you're not relating it to the effective diapycnal mixing? You're relating how resolving the wind stress curl and upper-ocean features to diapycnal fluxes of tracer properties within the Ekman layer, which has more to do with advection. It's possible, however, that, by conversation of volume in your model, diapycnal mixing will counter advective diapycnal fluxes but you are not showing this.

*A: There is a literature linking Ekman pumping to effective diapycnal transports. The idea probably being that Ekman pumping, for the very least, produces horizontal gradients which may be exposed to horizontal mixing and eventually translate to vertical transport. I am not in favor of this since our results do not show a link between Ekman transport and diapycnal transport (neither off Mauretania nor in the Baltic Sea: https://os.copernicus.org/articles/12/977/2016/ ). We brought this up since we felt that such a view is widespread in the literature. But you are right. We agree (after careful re-reading) that our presentation is somewhat odd here. We will either explain the alleged link between Ekman pumping and diapycnal mixing more comprehensively, or we will tone down the "Ekman story", or (most probably) do both in the revised version of the manuscript.*

R: Lines 313-319: The differences in domain-averaged Delta Ar[%] amounting to <0.1% with the same back diffusivities (equivalent to increasing the background diffusivity from 20% to 40%) may not be significant/detectable here once you consider the rate at which Delta Ar[%] and/or global kinetic energy is changing as the simulations are spinning-up. It appears that in Figure 7d that the seasonal variability and trend in (Delta?) Ar[%] may be larger than the difference across High and Coarse. So it's unclear whether the comparisons you make with the comparable effect of varying the background diffusivity and/or numerical advection scheme will hold up if your model was spun up more. A decrease in Delta Ar[%] from using a background diffusivity of 0.14 cm^2 s^-1 to 0.16 cm^2 s^-1 suggests varying rates of spin-up across different background diffusivities, for instance.

*A: Thanks for this comment. We realized that our method (especially the test and calibration simulations with altered background diffusivities) needs better explanation/presentation and further analysis to be convincing (maybe something like Figure 1 above). The fact that Figure 8 indeed does not show a strictly monotonic dependency between argon saturation*

*and background diffusivity suggests that there are, as we pointed out in the original version of the manuscript, antagonistic effects. Our interpretation is that the antagonistic effects limit the detection accuracy of our approach. As such we interpreted Figure 8 that the method generally works pretty well down to differentiating differences in diffusivities even down to almost ~2x10$^{-6}$ m$^2$/s - which we found sensational. That said, we understand that information on the temporal evolution of simulated Ar saturation in our simulations will make our approach more convincing. Also, we appreciate the earlier suggestion of the reviewer to exclude the (in this respect noisy and rather disturbing) mixed layer. We think something along Figure 1 (above) should be added to the revised version of the manuscript! (?)*

Minor corrections:

*A: Thank you for taking the time going through the text so carefully. We really appreciate this and (naturally) will correct everything accordingly.*

---

## Author Comment (AC2)

[Figure]

*Figure 1: Simulated time elapsed since water parcels have been in contact with the Northern, Southern or Western boundary of the model domain (monthly mean at the end of a 30 year-long integration of configuration "Coarse"). The time elapsed shown is averaged over the 100m to 400m depth interval (surface values are discarded following a suggestion by RC1). The red contour depicts the 3-year isoline indicating that the contoured region is rather undisturbed by boundary effects at the end of a 3 year-long simulation.*

---

## Author Response (AR1)

Kiel, December 5th, 2024

Subject: Resubmission of Manuscript: Mixing, Spatial Resolution, and Argon Saturation in a Suite of Coupled General Ocean Circulation Biogeochemical Model Configurations off Mauretania

Dear Editor,

We are pleased to resubmit our revised manuscript, incorporating substantial improvements in response to the very constructive feedback provided by the reviewers. Among the major updates to the manuscript following the reviewers' guidance are a more detailed presentation of the temporal evolution of argon saturation and an analysis of potentially spurious effects of our boundary conditions. The respective updates comprise additional figures and model simulations/analysis. Also, we introduced a new discussion part which discusses the precision of our method and puts our method now better into perspective to other approaches.

Please find the detailed point-by-point response to the reviewers' feedback below. The reviewer comments are marked in bold while our comments are in italics.

Thank you for your time and effort!

Kind regards,
        the authors

**Point by point response**
* * *
--------------------------------------------------- Reviewer #1    ---------------------------------------------------
* * *
*We are grateful to reviewer #1 who has obviously spent a lot of time with our manuscript, put the finger on shortcomings of our original presentation and made a lot of very constructive propositions as to how to become more convincing.*

**R: General comments:**
**The authors of this manuscript examine the model-simulated inaccuracies in diapycnal mixing that cannot be explained by the horizontal resolution near Mauretania and find that it's comparable to advection numerics and choices of background diffusivity by making use of argon saturation as a proxy for effective mixing. I can appreciate how much time and effort that went into this work but I have some reservations about publishing this manuscript as is. One concern that I have is that there is no comparison of the effective diapycnal mixing with other methods, like that of Holmes et al. (2021) or potentially separating out the spurious contribution to compare with a method like that developed in Ilıcak (2016)'s "Quantifying spatial distribution of spurious mixing in ocean**

models". I understand the authors just want to have a relative measure to rank their model configurations with their effective mixing against one another but without knowing whether their method is properly quantifying the effective mixing, the ranking could be misleading. For example, the authors include the mixed layer in their calculations of the effective mixing, despite how there are known confounding effects on the effective mixing. The second issue is that the authors try to connect Ekman pumping/suction to effective diapycnal mixing without explaining the relationship between the diapycnal advection and diapycnal mixing in the Ekman layer. The third issue is that the model spin-up time may suffice for many applications of model experiments, but they are investigating a signal that is so small that the rate at which their model diagnostics are changing at the time they are evaluating them may be relevant to the significance of the small differences they're finding. I suggest major revisions. Specific comments are listed below:

*A: Basically, the reviewer asks for a more convincing presentation of the capabilities of our new method along with a more comprehensive embedment of our work in that of others. This makes sense to us.*

*More specifically we find the reviewer's "issues 1 and 3" are essentially rooted in the same shortcoming of our original manuscript. As we understand, the perception of the reviewer is the following: (1) we present a new method to rank the effect of vertical diffusivity/diapycnal mixing, (2) then we find that our method is affected antagonistically by diffusivity, (3) we run the model/method only for a short period that appears random in length to the reader; such that the reader cannot rule out that the effects we discuss are also random - in the sense that we may have picked a time where the relation is coincidentally in favour of our method/reasoning and finally, (4) we do not make a good job to put the capabilities of our approach into the context of what is already out there.*

*We revised our manuscript considerably in order to overcome this shortcoming: in the revised version we added configuration Coarse2 to Figure 7 and introduced an additional Figure 8, showing the temporal evolution of all simulations throughout year two and three. Further we added rates to Table 2. A discussion section was added (starting ln. 330), stating that the saturation signals due to mixing are indeed small in amplitude - but nonetheless consistent over time. From this we derive an estimate on the precision of our method. The new discussion section does also put our results into the context of similar contemporary methods - as suggested by the reviewer.*

*As concerns "issue 2": the Ekman pumping has been put forward in the literature to make the point that resolution matters in terms of vertical (and eventually diapycnal) transports. We realize (and agree with the reviewer), however, that such a relation is neither obvious nor stringent. We changed the text accordingly and state now in line 285: "... We are particularly curious whether the increase in Ekman dynamics with increasing resolution leads to enhanced diapycnal mixing or if its effect is primarily advective. ..."*

*In summary, the new discussion Section addresses issue #1. Issue #2 is resolved in line 285. The additional Figure 8 and the update of Table 2 address issue #3. The additional appendix B provides another rationale for the chosen spin-up length and the model run time (issue #3).*

**R: Specific comments:**

**Line 3: The authors say "unacquainted vertical" by which they are referring to "diapycnal" but that way of describing diapycnal sounds more literary than it sounds like it has specificity; did the authors use AI to help them write this abstract?**

*A: We deleted "unacquainted".*

**R: Lines 38-39: The canonical reference for uncertainties in representation of diapycnal mixing (not just vertical) is MacKinnon et al. (2017)'s "Climate Process Team on Internal-Wave Driven Ocean Mixing" even though there's also shear-driven and other mixing**

*A: Thanks. We included the reference in ln. 39.*

**R: Line 45: This seems like more "coupled" reasoning than circular because the biogeochemical modules can potentially be developed to be more realistic without developing the diapycnal diffusivity modules, but of course the biogeochemical variables depend upon the diapycnal diffusivities. It's difficult to point to new model innovations in ESMs that don't have this coupling problem. Your argument seems analogous to saying that assimilating sea-ice concentrations with the goal of improving the sea surface properties is circular reasoning because our sea-ice modules are imperfect. In any case, without improvements in the biogeochemical modules, the information propagated by the assimilation of biogeochemical variables may not inform the diapycnal diffusivities as much as the biogeochemical parameters themselves, which is the problem you're pointing out. This would certainly be the case with the Green's function-based approach of ECCO-Darwin (see Carroll et al. (2020)'s "The ECCO-Darwin Data-assimilative Global Ocean Biogeochemistry Model: Estimates of Seasonal to Multi-decadal Surface Ocean pCO2 and Air-sea CO2 Flux") because the physical variables aren't changed in their formulation; only the biogeochemical parameters are altered. I believe the proposal by Trossman et al. (2022) is that the ECCO-Darwin and ECCO setups could be used in an iterative process to incrementally improve the biogeochemical parameters and the physical parameters, respectively, but this is even more computationally expensive than the current process and may require a different process for sequential data assimilation systems, which they present unique issues with in their appendix. Additional evidence that this could work comes from Skakala et al. (2022)'s "The impact of ocean biogeochemistry on physics and its consequences for modelling shelf seas," where it was shown that assimilation of a suite of biogeochemical variables had an impact on the diapycnal diffusivities beyond that of physical variables.**

*We removed the "circular reasoning" an toned the paragraph down so that it is no longer so pessimistic in that we merely state that the approach is not guaranteed to succeed in all cases (ln. 44 - 47).*

**R: Lines 63-64: Will this always be the case? The atmospheric community seems to have come up with a solution to advection numerics in McGraw et al. (2024)'s "Preserving Tracer Correlations in Moment-Based Atmospheric Transport Models". Also, is it possible that spurious diapycnal mixing is the result of spurious vertical advection sometimes? This certainly happens when you perform sequential assimilation (see: Pilo et al. (2018)'s "Impact of data assimilation on vertical velocities in an eddy resolving ocean model").**

*We included the McGraw (2024) reference in ln. 62 because we agree that it makes sense to point to ongoing developments. As to the applicability of their approach in ocean models (where all vertical layers are touching land somewhere in contrast to atmospheric models where most vertical layers are above orographic features) we are not so sure.*

*Yes, advection schemes can introduce spurious effects in all dimensions, including vertically. We added "... 3-dimensional ..." to ln. 60.*

**R: Lines 81-82: Isn't there also the effect of surface disequilibrium at the time of water mass formation (e.g., when subduction occurs) that needs to be accounted for? Both the nonlinearity of solubility in the presence of mixing and surface disequilibrium effect cause apparent oxygen utilization to be different from true oxygen utilization, for example. So the argon tracer needs to be injected below the mixed layer such that it doesn't get obducted throughout the experiment. Also, argon can't be injected into the ocean to derive diapycnal diffusivities from observations (unless the background levels are well-known) because there are significant background levels of argon in the ocean, which will eventually render it another exotic tracer on top of the ones Ledwell used. I mention this because it was noted that argon could be used as a proxy for effective diapycnal mixing in both models and observations in the text.**

*A: Agreed. We now stress this point more prominently in an additional Discussion Section (starting ln. 337).*

**R: Line 150: I understand not wanting to compete with Jason Momoa in Google searches by dubbing your model's acronym MOMOA, but you're going to compete with the Museum of Modern Art in New York City instead. In any case, the model and its tracer conservation properties won't be as much of a problem as they would be if the authors chose another model like HYCOM. And the model domain is reasonable, considering problems with boundary conditions that can occur.**

*A: The model has already been published under this name. As concerns HYCOM: We are really interested in testing our method in an isopycnal model!*

**R: Line 169: You say that there are 55 geopotential levels in all model configurations here but in Table 1, it says there are 72 grid points in an undefined dimension. The caption of Table 1 only mentions zonal and meridional directions but there is a third dimension listed for the number of grid points. Is that dimension time or is one of your instances of vertical dimensions inaccurate (i.e., in Table 1 or on Line 169)?**

*A: Thank you for catching this. Ln. 169 was in error and is corrected now (now ln. 170). There are 72 vertical boxes.*

**R: Line 186: For future equilibrations of biogeochemical variables, you could consider using another method like the one by Khatiwala (2024)'s "Efficient spin-up of Earth System Models using sequence acceleration"**

*A: We added the respective citation to the revised version of the manuscript (now ln. 188).*

**R: Line 189: I would argue that use of COREv2 forcing is no longer considered standard, as JRA55-do has supplanted COREv2 in OMIP2 and ERA5 is also commonly used**

*A: You are right. We added the references and changed the wording to: "... a well-tested annual climatological cycle of all the data needed to force an ocean model (note that similar contemporary products exist; Tsujino et al., 2018; Hersbach et al., 2020)." now ln. 190*

**R: Lines 201-203: If you're interested in the evolution of each of the model configurations, then you should be assessing metrics that are related to the time rate of change, not averages**

*A: We added rates of change to Table 2 and added the temporal evolution of simulation Coarse2 to Figure 7d. In addition, we show now the temporal evolutions of all simulations in the new Figure 8. We hope that this additional information makes our approach more convincing.*

**R: Lines 205-206: One year of spin-up is common amongst published high-resolution model simulations (based on the global kinetic energy metric you seem to be using) because the initial shock is essentially adjusted away but the model's climatological state is still being approached asymptotically. That would need at least several more years to achieve. It may take much longer to equilibrate potential energy, which includes available potential energy relevant to the eddying circulation, but to first order using kinetic energy may be good enough.**

*A: You are right. What we actually want to say is that a comparison of simulations without any energy would be meaningless, because without movement all resolutions will look alike. So having enough energy, i.e. not falling short of observational estimates, is essential for our study. But, as you pointed out rightfully, it does not prove that energy has equilibrated. We changed the text to (now ln. 204): " ... We compare simulations that have not reached equilibrium yet (after the short integration of only three years at most) which constrains potentially spurious effects of our lateral boundary conditions (c.f. Appendix \ref{app:boundary}) on our results and saves computational resources. Note in this context that \citet{dilma21} showed that the circulation of MOMA is already realistic after a spin-up of only one seasonal cycle. Consistent with this result we find that the energy content of our eddy-field does not fall short of observational estimates after one year already (Figure~\ref{fig:05}). This suggests that our simulation period of 3 years is long enough to*

*reveal major effects of differing resolution (even though it may take much longer to equilibrate potential energy). ..."*

**R: Table 2: Continuing on from my previous comments, please include the time rate of change (linear slope of a regression against time, for example) of the Delta Ar[%] metric you include the average of in this table. That would help the reader see whether these values are close enough to what they would be in a more well-spun-up state. Also, why are you including the mixed layer in your calculations when Argon is not a conservative tracer near the sea surface and diapycnal mixing is not very meaningful within a bulk mixed layer? Are you not using a bulk mixed layer?**

*A: We added respective rates to Table 2. Further, we added the simulation Coarse2 to the temporal evolution in Figure 7. In addition, we introduced Figure A1 which shows the temporal evolution of all simulations (with the exception of Coarse upwind which was off the scale). Further, we excluded the upper 100m from all of our analysis which, indeed, removes some of the noise. Thank you for the suggestion!*

**R: Lines 234-236: Again, because penetrating solar radiation, bubble entrainment, and possibly other processes affect Delta Ar[%], why are you including the mixed layer in your calculations?**

*A: You are right. By discarding the upper water column down to the maximum winter surface mixed layer we actually got rid of a lot of noise (mainly the effect of subsurface solar heating accumulating below the summer surface mixed layer). We changed Table 2 and Figure 7d accordingly.*

**R: Lines 266-268: This is qualitatively accurate. However, it can be noted that there are in-situ estimates of eddy kinetic energy compared with high-resolution model simulations in the literature, like those presented in Luecke et al. (2020)'s "Statistical Comparisons of Temperature Variance and Kinetic Energy in Global Ocean Models and Observations: Results From Mesoscale to Internal Wave Frequencies". Also, along-track SWOT data resolves the higher wavenumbers and there are preliminary results that much more of the spectrum is resolved with SWOT. I'm not suggesting that you perform a comparison with in-situ (e.g., moored) or SWOT data. This is just something to be aware of.**

*A: Thanks. We added the respective reference (now ln. 269).*

**R: Figure 3: It's curious that the relatively warm SST feature near 18W, 21N in the High configuration is completely absent in Coarse, except for what seems like a separate merging feature near the upper part of the domain plotted here. The rest of the figure comparison looks like the High configuration's SST was coarse-grained.**

*A: Agreed. We changed the wording to (now ln. 251): "... On these scales it is evident that a higher spatial model resolution features more small-scale circulation features such as the relatively warm SST feature near $18^\circ$W, $21^\circ$N that is completely absent in Coarse. ..."*

**R: Table 3 and Figure 6: Okay, the higher the resolution, the greater the Ekman pumping/suction there is in its mean and variability, but how come you're spending time on this when you're not relating it to the effective diapycnal mixing? You're relating how resolving the wind stress curl and upper-ocean features to diapycnal fluxes of tracer properties within the Ekman layer, which has more to do with advection. It's possible, however, that, by conversation of volume in your model, diapycnal mixing will counter advective diapycnal fluxes but you are not showing this.**

*A: You are right. We changed the wording to (now ln. 283): " Hence we expect, consistent with the pioneering work of, e.g., \citet{levy01, mahadevan00, mahadevan06, martin03}, an increase in vertical, fluxes of heat, salt and biogeochemical entities with increasing resolution in our configurations. We are particularly curious whether the increase in Ekman dynamics with increasing resolution leads to enhanced diapycnal mixing or if its effect is primarily advective.*

**R: Lines 313-319: The differences in domain-averaged Delta Ar[%] amounting to <0.1% with the same back diffusivities (equivalent to increasing the background diffusivity from 20% to 40%) may not be significant/detectable here once you consider the rate at which Delta Ar[%] and/or global kinetic energy is changing as the simulations are spinning-up. It appears that in Figure 7d that the seasonal variability and trend in (Delta?) Ar[%] may be larger than the difference across High and Coarse. So it's unclear whether the comparisons you make with the comparable effect of varying the background diffusivity and/or numerical advection scheme will hold up if your model was spun up more. A decrease in Delta Ar[%] from using a background diffusivity of 0.14 cm^2 s^-1 to 0.16 cm^2 s^-1 suggests varying rates of spin-up across different background diffusivities, for instance.**

*A: We added the Coarse2 simulation to Figure 7d and rates of change to Table 2.  Also we introduce the additional Figure 8 which shows the temporal evolution of Ar saturation in all of our simulations. Our argument is that, although the signals are rather small, they are distinguishable and consistent over (quite some) time.*

**Minor corrections:**
A: Thank you for taking the time going through the text so carefully. We really appreciate this and corrected everything accordingly.
* * *
------------------------------------------------    Reviewer #2    --------------------------------------------
* * *
We are grateful to reviewer #2! Her/his issue with potentially spurious effects of boundary effects is justified and we think that the manuscript has become stronger now that we added a respective analysis based on a simulation of the time elapsed since water parcels were in contact with the boundaries.

**R: Summary:**

**The main finding of this paper is that diapycnal mixing, a key source of uncertainty in ocean models, is strongly influenced by factors such as model resolution, advection schemes, and background diffusivity. Using Argon saturation as a proxy, the study shows that the effect of differences in resolution is comparable to that of other factors. This study highlights the ongoing challenge of accurately representing ocean mixing in models used for climate projections. I found this manuscript concise and well-written. However, some clarifications in methods and a few minor corrections are needed before publication. I suggest minor revisions. Please see the general comments and specific comments below.**

*A: Thank you. We found your comments very helpful, especially those related to the impact of the boundaries. We have added respective additional analysis & discussion (discussion ln. 330ff, Appendix B: Boundary Conditions ln. 447ff) and think the paper has become stronger.*

**R: General comments:**

**1.   Consider changing the title to highlight the key findings or purpose of this study. The current title is concise, but it could be refined for better focus and impact.**

*We changed the title to: "Mixing, Spatial Resolution and Argon Saturation in a Suite of Coupled General Ocean Circulation Biogeochemical Model Configurations off Mauretania."*

**2.   While the rationale for using a 3-year simulation period is clear in terms of model spin-up and avoiding equilibrium, I have concerns about the potential influence of spurious effects from the closed boundaries on the region of interest. Could the authors provide diagnostics or an analysis that evaluates how far spurious effects, such as reflected waves or boundary-induced disturbances, propagate into the domain over the 3-year period? This would help confirm whether the region of interest remains unaffected.**

*A: We added a new analysis to the appendix (starting in now ln. 443) showing that it takes, on average, more than three years until material (as opposed to energy) is transferred from the boundaries into the high-resolution domain. In ln. 204 we state now: "... We compare simulations that have not reached equilibrium yet (after the short integration of only three years at most) which constrains potentially spurious effects of our lateral boundary conditions (c.f. Appendix \ref{app:boundary}) on our results and saves computational resources. ..."*

**R: Detailed comments:**

**1.  Line 134: What is the horizontal resolution of MOMSO? Is it 50 km as shown in Fig. 1?**
*It is less than 11km. We added this information to ln. 134.*

**2.  Line 165: "resolutions" should be "resolution."**
A: Yes, thank you.

**3.  Line 169: I have a question regarding the vertical grid description in line 169, where it is mentioned that the model uses 55 vertical levels but 72 vertical grid points in Table 1. Could you kindly clarify this aspect? Is the difference in grid points due to the use of staggered grids, boundary conditions, or another feature of the model's vertical discretization? If there is a specific reason for this configuration, I would appreciate further details as a reader.**
*A: Sorry. Ln. 169 is wrong. There are 72 vertical levels. Corrected. Now ln. 170.*

**4.  Figure 1 caption: Panel (c) and panel (d) don't show vertical resolution, right? Am I missing anything?**
*A: You are correct. We deleted "vertical" in the last sentence of the caption.*

**5.  Line 188: I believe "affect" would be better than "effect" here.**
*A: We replaced "affect" by "drive"; now ln. 190*

**6.  Line 225: It seems to me that equation (1) is incorrect. Since delta_Ar = Ar - Ar_sat, why do we need to subtract 100% in this equation? This might be a typo, but if not, please check the calculations presented in this study. Or am I missing anything?**
*A: No, you are right! This is a typo. Corrected. Now in ln. 226.*

**7.  Line 229: Could the authors elaborate on why mixing always causes an increase in saturation? Wouldn't the sign of the change in saturation depend on the temperature before and after mixing, given that Argon saturation is a non-linear function of temperature?**
*A: You are right, non-linearity is not enough. We changed the wording to: " ... In contrast to concentration the saturation is not conservative: a mixed water-parcel will always carry a higher argon saturation than the arithmetic mean of the original parcels since the saturation curves are convex over the entire range of oceanic temperatures (and salinities). Hence, an increase in $\Delta Ar$\% may be indicative of mixing. ..." Now in ln. 226.*

**8.  Line 229: There is a typo. Change "his" to "is."**
A: Fixed.

**9. Line 233: "Allows" should be "allow."**
*A: Fixed.*

**10.  Line 266: There is an extra parenthesis.**
*A: Fixed.*

**11.  Line 295: "loses" should replace "looses."**
*A: Fixed.*

---

## Referee Report (RR1)

Review #2 of Dietze and Löptien's "Argon Saturation in a Suite of Coupled General Ocean Circulation Biogeochemical Models off Mauretania" (manuscript #: 2024-918)

General comments:

The authors of this manuscript examine the model-simulated inaccuracies in diapycnal mixing that cannot be explained by the horizontal resolution near Mauretania and find that it's comparable to advection numerics and choices of background diffusivity by making use of argon saturation as a proxy for effective mixing. The method makes use of Argo saturation and is simpler than previous methods for detecting spurious diapycnal mixing, except is somewhat similar (in signal detection capability and complexity) to that of Holmes et al. (2021). The distinction between using Argon saturation for diagnostic purposes (spurious diapycnal mixing) as opposed to constraining mixing parameters in ocean models in a data assimilation application needs to be better clarified because there are long passages in the introduction's text that seem to suggest that the authors are going to choose Argo saturation over, for example, biogeochemical tracers, which are used (e.g., in a submitted paper by Ellison et al.) for the other application. This manuscript further points out nuances with using Argo saturation as a proxy for diapycnal mixing, such as the fact that equilibrating air-sea fluxes can erase oversaturation signals, which makes Argo saturation less than perfect for diagnosing spurious diapycnal mixing. But in the way the authors experiments are designed (e.g., using different background diffusivities), their conclusions can be drawn. I suggest technical revisions. Specific comments are listed below:

Specific comments:

Line 21: "erroneous" is still spelled incorrectly (currently spelled "erroeneous")

Line 68: "… so complex that the essence…" should have an exampled cited such as "… so complex (e.g., Ilıcak (2016)) that the essence…" where Ilıcak (2016) is Ilıcak's "Quantifying spatial distribution of spurious mixing in ocean models" in Ocean Modelling

Line 81: Before the sentence, "The relatively novel approach…" the authors should insert a sentence like, "While argon saturation is poorly observed, especially when compared with the observational coverage of biogeochemical tracers, for example, this study makes use of argon saturation as a model diagnostic, not a data constraint." I suggest this because the reader may get the idea that Argo saturation is used as a data constraint to calculate ocean mixing parameters with the previous discussion about how studies have done that to date, and if one wants to ultimately use observations of Argo saturation to guide ocean mixing, they will only have a single transect going south from Alaska, a couple of zonal transects in the equatorial Atlantic Ocean, and a couple of meridional transects in the equatorial Atlantic Ocean of Argo concentrations in the World Ocean Database. This seems like the best place to clarify that Argon saturation is used here more for diagnostic purposes of spurious diapycnal mixing in ocean models.

Line 205: "… integration time of maximal three years) which constrains potentially spurious effects…" should be "… integration time of a maximum of three years) which limits potentially spurious effects…"

Line 230: "…can be indicative for spurious dispersion)." should be "… can be indicative of spurious dispersion)."

Lines 335-336: The sentence, "We suspect that the same holds…" can be deleted because it doesn't add anything (just speculation).

Line 393: The authors are missing an end parenthesis: "(Porcupine Abyssal Plain and POMME region, respectively)."